# Scheduled feeding improves behavioral outcomes and reduces inflammation in a mouse model of fragile X syndrome

Huei-Bin Wang[1,2], Natalie E Smale[2], Sarah H Brown[2], Sophia AMB Villanueva[2,3], David Zhou[2], Aly Mulji[3], Deap S Bhandal[2], Kyle Nguyen-Ngo[2], John R Harvey[2,3], Cristina A Ghiani[2,4]*, Christopher S Colwell[2]*

[1]Molecular, Cellular, Integrative Physiology Graduate Program, University of California, Los Angeles, Los Angeles, United States; [2]Department of Psychiatry & Biobehavioral Sciences, University of California, Los Angeles, Los Angeles, United States; [3]Integrated Biology and Physiology Program, University of California, Los Angeles, Los Angeles, United States; [4]Department of Pathology and Laboratory Medicine, David Geffen School of Medicine; University of California, Los Angeles, Los Angeles, United States

*For correspondence:
cghiani@mednet.ucla.edu (CAG);
CColwell@mednet.ucla.edu
(CSC)

Competing interest: The authors declare that no competing interests exist.

## eLife Assessment

This manuscript presents **solid** experimental data using Fmr1 knockout mice to explore the **fundamental** role of Fmr1 in sleep regulation. The study supports the hypothesis that scheduled feeding can improve circadian rhythm and behavior in a mouse model of Fragile X syndrome. These findings may offer new insights into neurodevelopmental disorders and their potential treatment strategies.

**Abstract** Fragile X syndrome (FXS), a leading inherited cause of intellectual disability and autism, is frequently accompanied by sleep and circadian rhythm disturbances. In this study, we comprehensively characterized these disruptions and evaluated the therapeutic potential of a circadian-based intervention in the fragile X mental retardation 1 (*Fmr1*) knockout (KO) mouse. The *Fmr1* KO mice exhibited fragmented sleep, impaired locomotor rhythmicity, and attenuated behavioral responses to light, linked to an abnormal retinal innervation and reduction of light-evoked neuronal activation in the suprachiasmatic nucleus. Behavioral testing revealed significant deficits in social memory and increased repetitive behaviors in the mutants, which correlated with sleep fragmentation. Remarkably, a scheduled feeding paradigm (6 hr feeding/18 hr fasting) significantly enhanced circadian rhythmicity, consolidated sleep, and improved social deficits and repetitive behaviors in the *Fmr1* KO mice. This intervention also normalized the elevated levels of some pro-inflammatory cytokines, including IL-12 and IFN-γ, in the mutants' blood, suggesting that its benefits extend to inflammatory pathways. These findings highlight the interplay between circadian disruption, behavior and an inflammatory response in FXS, and provide compelling evidence that time-restricted feeding may serve as a promising non-pharmacological approach for improving core symptoms in neurodevelopmental disorders.

## Introduction

Fragile X syndrome (FXS) is a relatively common inherited cause of intellectual disability, with a prevalence of approximately 1 in 4000 males and 1 in 8000 females (*Hersh et al., 2011*). Males are typically

more severely affected due to their single X chromosome. FXS is caused by an abnormal expansion of CGG trinucleotide repeats in the *fragile X mental retardation 1* (*FMR1*) gene, which results in the transcriptional silencing of *FMR1* gene and consequent reduction in the levels of the fragile X mental retardation protein (FMRP), an RNA-binding protein essential for synaptic plasticity and neuronal cytoarchitecture (*Zalfa et al., 2006*; *Soden and Chen, 2010*; *Kute et al., 2019*). There is currently no cure for FXS. Existing treatments are primarily symptomatic, targeting common manifestations such as social deficits, repetitive behaviors, and sleep disturbances (*Budimirovic et al., 2022*; *Martinez et al., 2024*). Given the essential role of sleep and circadian rhythms in neural recovery and plasticity, a compelling question is whether improving sleep/wake cycles might alleviate other core symptoms of this disease.

Animal models are invaluable for elucidating the mechanisms underlying circadian and sleep disturbances as well as developing interventions aimed at restoring healthy sleep/wake cycles. Among the available models for FXS, the *Fmr1* knockout (KO) mouse has been extensively validated for investigating disease mechanisms and is widely used in preclinical drug development, including for sleep-related outcomes (*Thomas et al., 2012*; *Kazdoba et al., 2014*; *Kat et al., 2022*; *Saré et al., 2022*; *Martinez et al., 2024*). Early studies have shown disrupted rhythms in activity during the light/dark (LD) cycle in mice missing both the *Fmr1* gene and the paralog *Fxr2* (*Zhang et al., 2008*). More recent findings indicate impairments in both sleep (*Saré et al., 2017*) and activity rhythms (*Bonasera et al., 2017*; *Angelakos et al., 2019*) in *Fmr1* KO mice, all consistent with the sleep/wake disturbances reported in FXS individuals (*Kronk et al., 2009*; *Kronk et al., 2010*; *Budimirovic et al., 2022*).

Despite these findings, significant gaps remain in our understanding of the precise nature of the sleep and circadian deficits in *Fmr1* KO mice, and whether targeted correction of these rhythms might mitigate behavioral symptoms. Furthermore, much of the behavioral research in *Fmr1* KO mice has been conducted during the light phase—when mice are typically asleep—potentially confounding results due to sleep disruption.

In this study, we aimed at comprehensively characterizing sleep and circadian disturbances in the *Fmr1* KO mice and examining their association with behavioral alterations. To avoid disrupting natural sleep patterns, we employed non-invasive home-cage monitoring systems to assess sleep and activity rhythms and conducted behavioral testing during the active (dark) phase. Importantly, the same animals were evaluated for both sleep/wake cycles and behavioral phenotypes, enabling direct correlations. Finally, because of the above-mentioned evidence of circadian dysfunction, we tested the efficacy of a circadian-based intervention—scheduled feeding (6 hr feeding/18 hr fasting)—on improving sleep/wake rhythms, social behavior, and repetitive behaviors in these mutants.

## Results

### Fragmented light-phase sleep in the *Fmr1* KO mice

To determine whether the *Fmr1* KO mice exhibit deficits in immobility-defined sleep behavior, the animals were examined using a combination of video recording and mouse tracking system under a 12:12 hr LD condition. Both the WT and mutant mice exhibited robust circadian rhythms in sleep with higher levels during the day and minimal sleep during the early dark phase (*Figure 1A*). The amount of total sleep within the 24 hr cycle and the number of sleep bouts did not differ between genotypes, although the *Fmr1* KO did exhibit a reduction in the average duration of each sleep bout (*Table 1*). A two-way ANOVA used to analyze the temporal pattern of sleep (1 hr bins) in WT and mutants revealed significant effects of time ($F_{(23, 287)} = 31.94$; $p < 0.001$) and genotype ($F_{(23, 287)} = 11.95$; $p < 0.001$), along with differences in day and night sleep parameters (*Table 1*). During the resting phase (lights on), the *Fmr1* KO mutants slept significantly less (*Figure 1B*) with more frequent sleep bouts (*Figure 1C*) of shorter duration (*Figure 1D*). The maximum sleep bout duration was about 20 min shorter in the mutant mice. The reduced sleep time, shorter bout length, and greater number of sleep bouts seen during the day are all evidence of a fragmented sleep pattern in the *Fmr1* KO mutants. The negative findings should be viewed with caution as the sample size of $n = 6$ per group is below ideal.

### Disrupted diurnal and circadian activity rhythms in the *Fmr1* KO mice

Next, we investigated the presence of deficits in locomotor activity rhythms in the *Fmr1* KO mice. Age-matched WT and *Fmr1* KO were housed in cages equipped with running wheels under 12:12 hr

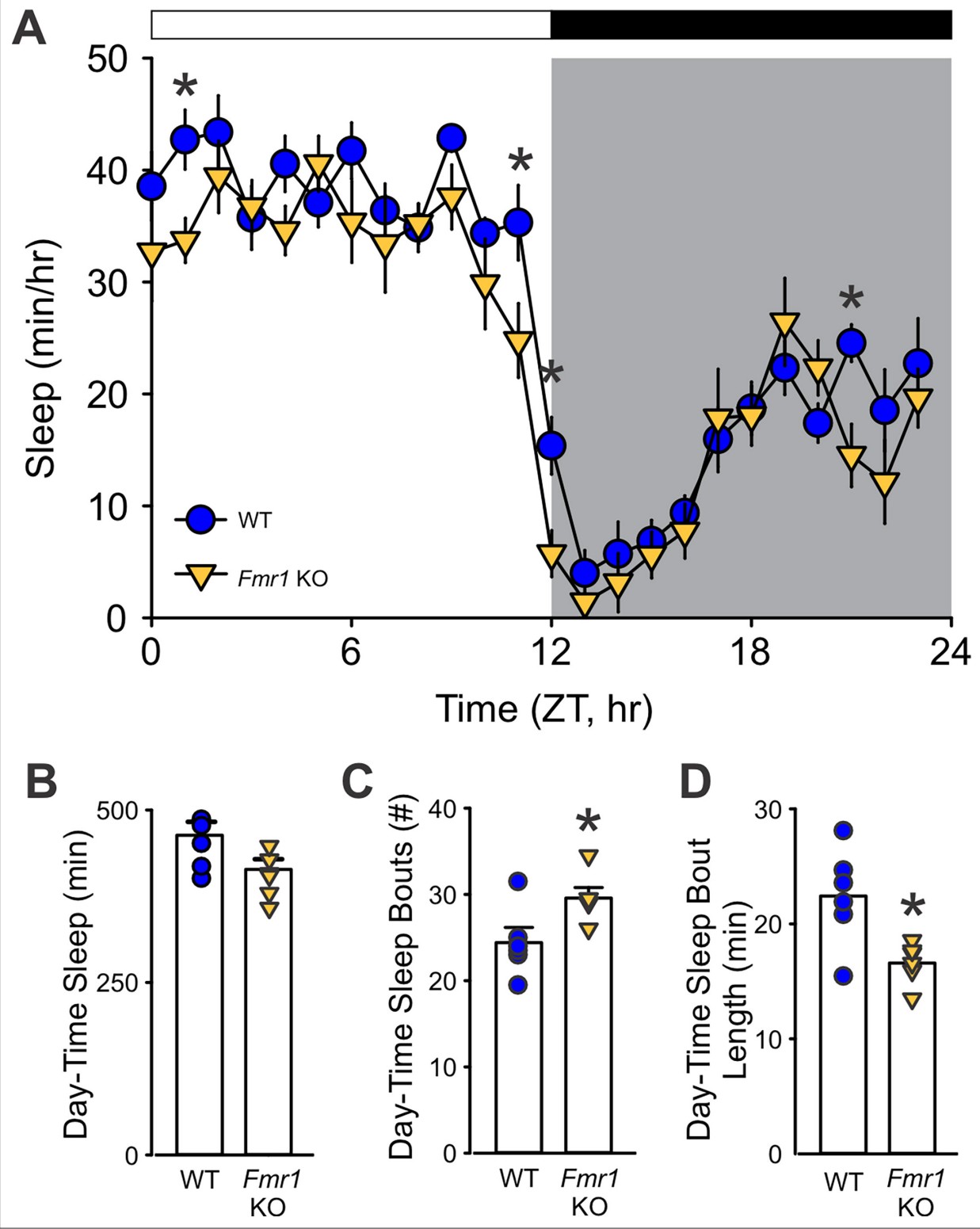

**Figure 1.** The *Fmr1* KO mice exhibit reduced and fragmented sleep during the light phase. (**A**) Waveforms of daily rhythms in sleep behavior under standard 12:12 hr light–dark (LD) cycles in WT (blue circle) and *Fmr1* KO (yellow triangle) mice. Zeitgeber Time (ZT) 0 corresponds to lights on, and ZT12 to lights off. Sleep was defined by immobility and binned in 1-hr intervals. Both genotypes exhibited clear diurnal rhythms, but the *Fmr1* KO mice showed significantly less sleep during the light phase and at several times in the dark phase. Asterisks indicate significant differences between genotypes at individual time points (two-way ANOVA with genotype and time as factors, followed by the Holm–Sidak's multiple comparisons test, *p < 0.05). The white/black bar on the top indicates the LD cycle; gray shading indicates the dark-phase time-period. (**B–D**) Immobility-defined sleep metrics

*Figure 1 continued on next page*

*Figure 1 continued*

during the light-phase. Compared to WT, *Fmr1* KO mice showed a greater number of sleep bouts, indicating sleep fragmentation, and of shorter duration. Histograms show the means ± SEM with the values from individual animals overlaid. Genotypic differences were analyzed by *t*-test (*p < 0.05). See **Table 1**.

LD cycles for 2 weeks and then released into constant darkness (DD) to measure their endogenous rhythms (**Figure 2—figure supplement 1**). Both genotypes exhibited robust daily and circadian rhythms in wheel running activity (**Figure 2A** and **Table 2**). Some indication of the commonly reported hyperactivity in the *Fmr1* KO mice (**Sørensen et al., 2015**) emerged in the LD cycle (**Figure 2B**), although the activity levels over 24 hr did not differ significantly (**Figure 2C**) between genotypes. Analysis of the diurnal activity patterns with two-way ANOVA confirmed significant effects of time ($F_{(23, 287)}$ = 8.84; p = 0.003) and genotype ($F_{(23, 287)}$ = 39.75; p < 0.001) on the locomotor activity. In LD (**Figure 2C**), the mutants exhibited lower rhythm power with increased activity during lights-on and more variability in the activity onset. When released in DD (**Figure 2D**), WT and mutants displayed very similar free-running period (tau), but again, the latter presented with weaker rhythm power and a higher cycle-to-cycle variability in activity onset (**Table 2**). Together, our data suggest that the *Fmr1* KO mice exhibit deficits in the circadian regulation of locomotor activity and raise the possibility of impaired ability to synchronize to the light cycle.

## Impaired behavioral response to photic cues in the *Fmr1* KO mice

The likelihood of the *Fmr1* KO mice presenting deficits in their response to light was tested using four behavioral assays of the retinal input to the circadian system. First, the ability of exposure to light (4500 K, 50 lx) during the dark phase to suppress or 'mask' the locomotor activity of the nocturnal animals was evaluated. For this assay, the level of locomotor activity between Zeitgeber Time (ZT) 14 and 15 (2 hr after lights off) was first measured under baseline LD conditions (i.e. the day before), and no differences were observed between genotypes (WT: 2229 ± 162 revolutions per hour (rev/hr); KO: 2639 ± 328 rev/hr; $t_{(18)}$ = −1.123, p = 0.277). The next day, the mice were exposed to light for 1 hr at this same phase, and the levels of activity measured. The mutants exhibited a significantly reduced light-driven suppression of the locomotor activity as compared to WT (**Figure 3A, B** and **Table 3**), with significant effects of both genotype ($F_{(1, 39)}$ = 12.002; p = 0.001, two-way ANOVA) and light exposure ($F_{(1, 39)}$ = 24.253; p < 0.001, two-way ANOVA). Next, we determined the number of cycles needed by the mice to re-entrain to a 6-hr phase advance of the LD cycle. While the WT mice re-synchronized in 5.9 ± 0.6 days to the new phase, the *Fmr1* KO mice required 11.3 ± 0.4 days to adjust to the new ZT 12 (**Figure 3C, D** and **Table 3**). Third, we evaluated the ability of the mice to entrain to a skeleton photoperiod (SPP) in which the full 12 hr of light or dark are replaced by two 1-hr light exposures (ZT 0–1 and ZT 11–12) separated by 11 hr of dark (**Figure 4A**). Mice were first entrained to the standard LD cycle and then released into the SPP for 2 weeks (**Figure 4A**). The activity recordings showed that the WT stably entrained to this challenging environment and exhibited robust circadian rhythms with a period (tau) of 24.0 ± 0.0 hr (**Table 3**). Conversely, the *Fmr1* KO exhibited a tau of 23.7 ± 0.2 hr with the shorter period driven by three mutants that failed to entrain to the SPP (**Table 3**). Compared to the WT, the mutants also showed reduced power of the rhythms, increased light-phase activity, and a larger onset variability (**Figure 4B**). Finally, the direct light-induced phase shift of the circadian system of the mice was measured by exposing them to light (300 lx) for 15 min while in constant darkness at circadian time (CT) 16. The WT showed a phase delay of 135.6 ± 26.9 min, while, in the *Fmr1* KO mutants, the delay in the activity onset on the next day was about half the magnitude (64.0 ± 0.1 min; **Figure 4C, D**; **Table 3**). These findings provide evidence for the presence of a defective circadian response to light in the *Fmr1* KO mice.

## Subtle deficits in the retinal afferent innervation to the suprachiasmatic nucleus in the *Fmr1* KO mice

The results described above suggest that the *Fmr1* KO mice present with an anomalous retinal input to the suprachiasmatic nucleus (SCN), the master circadian clock located in the anterior hypothalamus. Hence, to assess the integrity of this pathway, the WT and *Fmr1* KO mice received a bilateral intravitreal injection of the fluorescence-conjugated neurotracer Cholera Toxin (β subunit), previously used to map the projections of the melanopsin-expressing intrinsically photoreceptive retinal ganglion cells

**Table 1.** Altered behavioral sleep parameters in the *Fmr1* KO mice.

Comparisons of sleep behavior in age-matched male WT and *Fmr1* KO mice (*n* = 6/group). Values are shown as the averages ± SEM. For the 24 hr dataset, values were analyzed using a *t*-test. Possible day/night differences were analyzed with two-way ANOVA using genotype (WT vs. *Fmr1* KO) and time (day vs. night) as factors, followed by the Holm–Sidak's multiple comparisons test. Asterisks indicate significant differences between genotypes, while crosshatch those between day and night. Alpha = 0.05. Degrees of freedom are reported between parentheses. Bold values indicate statistically significant differences.

| 24 hr totals | | | WT | | Fmr1 KO | | t test |
|---|---|---|---|---|---|---|---|
| Sleep duration (min) | | | 646.1 ± 30.2 | | 568.6 ± 29.8 | | $t(10)$=1.973; p=0.077 |
| Sleep bouts (#) | | | 49.9 ± 2.7 | | 50.3 ± 2.0 | | $t(10)$=−0.136; p=0.895 |
| Sleep bout length (min) | | | 15.7 ± 1.1 | | 13.1 ± 0.6* | | **$t(10)$=2.329; p=0.042** |

| | WT | | Fmr1 KO | | Two-way ANOVA | | |
|---|---|---|---|---|---|---|---|
| Measures | Day | Night | Day | Night | Genotype | Time | Interaction |
| Sleep duration (min) | **464 ± 19.4#** | 182 ± 13.5 | **414 ± 14.8*#** | 154.6 ± 17.7 | $F(1,23)$=6.37; p=0.02 | $F(1,23)$=319.5; p<0.001 | $F(1,23)$=0.56; p=0.46 |
| Sleep bout (#) | 24.4 ± 1.8 | 25.5 ± 2.1 | **29.6 ± 1.2*#** | 20.7 ± 1.9 | $F(1,23)$=0.017; p=0.90 | $F(1,23)$=5.70; p=0.027 | $F(1,23)$=9.33; p=0.006 |
| Sleep bout length (min) | **22.4 ± 1.9#** | 9.0 ± 0.7 | **16.6 ± 0.8*#** | 9.5 ± 0.6 | $F(1,23)$=6.77; p=0.017 | $F(1,23)$=100.9; p<0.001 | $F(1,23)$=9.64; p=0.006 |
| Max bout length (min) | **81.3 ± 6.0#** | 30.5 ± 3.0 | **58 ± 3.09*#** | 38.7 ± 3.8 | $F(1,23)$=4.02; p=0.059 | $F(1,23)$=85.96; p<0.001 | $F(1,23)$=17.33; p<0.001 |

(ipRGCs) from the retina to the SCN (*Muscat et al., 2003*; *Hattar et al., 2006*). At variance with the WT, a lower fluorescent signal could be observed in the mutant mice both laterally and medially to the ventral SCN (*Figure 5A*), where the retino-hypothalamic fibers reach the nuclei, as well as beneath in the optic chiasm. Analysis of the intensity and distribution (*Figure 5—figure supplement 1*) of the labeled retino-hypothalamic processes reaching and entering the ventral SCN showed a reduction, particularly evident in the lateral side of both the left and right mutant SCN (*Figure 5B, C*). Likewise, a subtle decrease in the intensity of the labeled fibers was found in the whole SCN (*Table 4*) of the *Fmr1* KO mice as compared to WT**.**

A well-established test of the light input to the circadian system is the light-evoked cFos response in the SCN. In line with the results obtained with the Cholera Toxin suggesting an impaired light pathway to the SCN in the mutants, the number of cFos positive cells in the SCN of the *Fmr1* KO was greatly reduced (50%) in comparison to the WT mice (*Figure 5D, E* and *Table 4*). Contrary to other models of neurodevelopmental disabilities (NDDs; *Li et al., 2015*; *Lee et al., 2018*), the *Fmr1 KO* mice did not display any histomorphometrical alteration of the SCN (*Table 4*). Based on these findings, we can surmise the presence of a compromised connectivity between the retina and the SCN in the mutants, which could provide, at least in part, a mechanism for their difficulty in responding to photic cues.

## Social and repetitive behavior deficits in *Fmr1* KO mice

Social deficits and stereotypic symptoms are hallmark problems in autism spectrum disorders, which we sought to evaluate during the active (dark) phase, between ZT 16–18, in the two genotypes. Social behavior was tested with the three-chamber and the five-trial social interaction tests (*Figure 6*). In the first stage of the three-chamber test, both genotypes showed more interest toward the stranger mouse when given the choice between an inert object and a mouse. The direct comparison of the time spent with the object and the mouse did not show significant differences between the genotypes (*Table 5*). On the other hand, measurement of the social preference index (SPI) indicated that the *Fmr1* KO mice exhibited a reduced interest in the conspecific compared to WT (*Figure 6A* and *Table 5*). In the second testing stage, when the lifeless object was replaced with a second novel mouse and the first stranger became the familiar mouse (*Figure 6B*), the mutants spent more time with the familiar mouse than the WT (*Table 5*). Again, the social novelty preference index (SNPI) confirmed that the *Fmr1* KO exhibited reduced interest in novel mouse compared to WT mice (*Figure 6B* and *Table 5*). The reduced time spent exploring the novel-mouse chamber suggests that the mutants were, perhaps, unable to distinguish the second novel mouse from the first, now familiar, mouse,

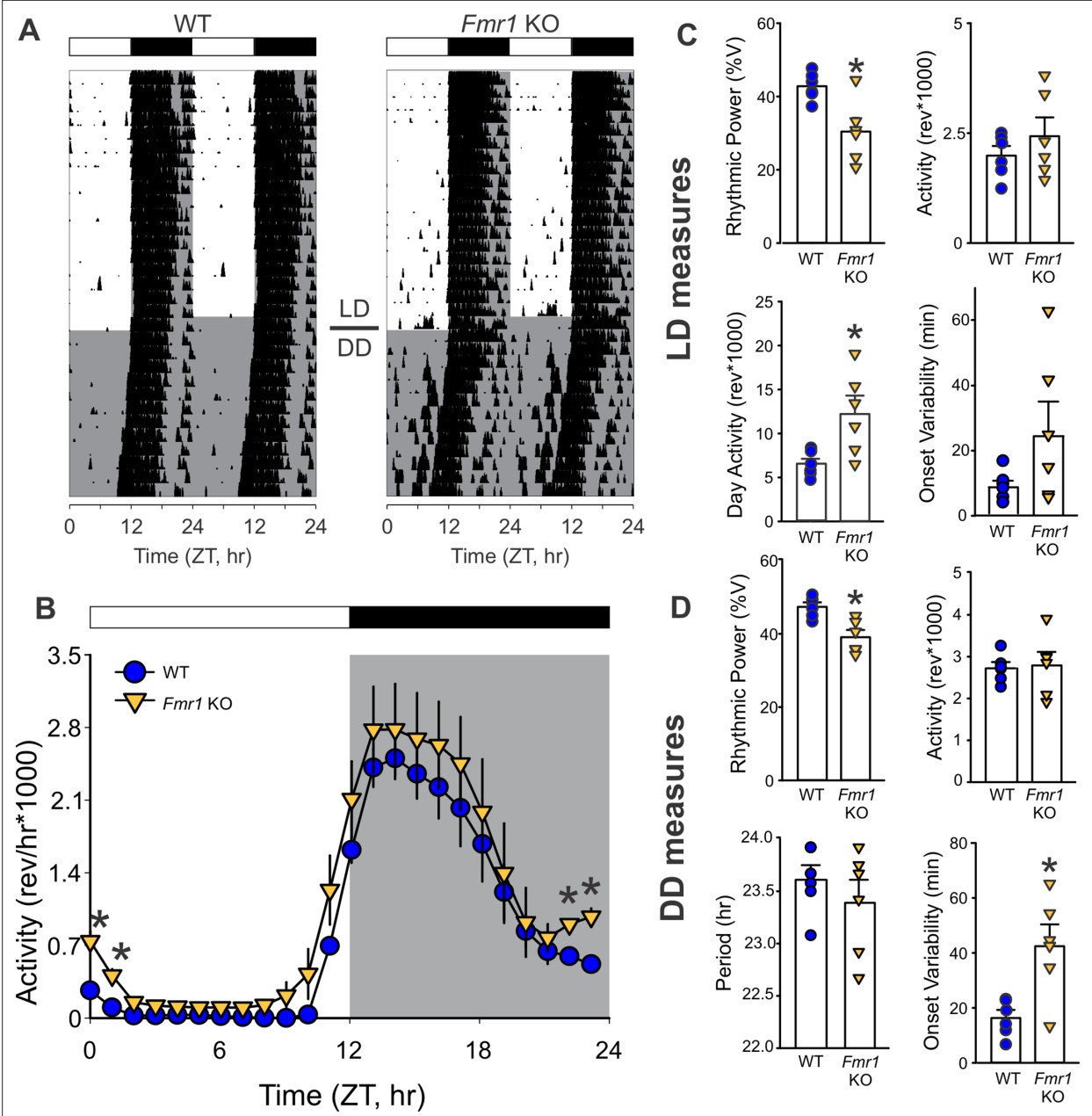

**Figure 2.** The *Fmr1* KO mice show unstable locomotor rhythms. (**A**) Representative actograms showing daily rhythms in wheel-running activity under LD followed by constant darkness (DD) in WT (left) and *Fmr1* KO (right) mice. Activity levels were normalized to 85% of the most active mouse. Each row represents two consecutive days, and the second day is repeated at the beginning of the next row. (**B**) Waveforms of daily rhythms in cage activity in WT (blue circle) and *Fmr1* KO (yellow triangle) mice under the LD cycles. Activity under LD (1 hr bins) was analyzed by two-way ANOVA with genotype and time as factors followed by the Holm–Sidak's multiple comparisons test (*p < 0.05). There were significant effects of both time (F = 8.84; p = 0.003) and genotype (F = 39.75; p < 0.001) on the temporal pattern of the locomotor activity rhythms. Note that significant genotypic differences were found before and after drawn. Activity metrics in LD (**C**) and DD (**D**). Rhythmic power under both LD and DD conditions was significantly reduced in the mutants, which also presented higher imprecision in the activity onset in DD. Histograms show the means ± SEM with the values from individual animals overlaid. Genotypic differences were analyzed by *t*-test (*p < 0.05). See *Table 2*.

The online version of this article includes the following figure supplement(s) for figure 2:

**Figure supplement 1.** Graphic presentation of the experimental design.

**Table 2.** Activity rhythms were altered in the *Fmr1* KO mutants.

The locomotor activity rhythms of adult male WT and *Fmr1* KO mice in the standard 12:12 hr LD cycles and constant darkness (DD) were monitored using wheel running activity (*n* = 6/group). Values are shown as the averages ± SEM. If the assumptions of normality and equal variance were met, a *t*-test was used to analyze the data; otherwise, the Mann–Whitney rank sum test was used. Asterisks indicate significant differences between genotypes. Alpha = 0.05. Degrees of freedom are reported between parentheses. Bold values indicate statistically significant differences.

| | WT | Fmr1 KO | Statistical values |
|---|---|---|---|
| **LD** | | | |
| Rhythmic power (% variation) | 42.8 ± 1.7 | **30.4±3.8*** | *t*(10)=3.309; p=0.008 |
| Cage activity (rev) | 19,875 ± 2197 | 24,302 ± 4294 | *t*(10)=−1.005; p=0.340 |
| Amplitude (rev) | 2553 ± 246 | 2771±406 | *t*(10)=−0.502; p=0.630 |
| Activity in the light phase (ZT 0–3, rev) | 418 ± 50.7 | **1099 ± 165*** | **U=0.000; p=0.002** |
| Onset variability (min) | 8.1 ± 2.0 | 24.5 ± 10.6 | *t*(10)=−2.060; p=0.066 |
| Fragmentation (bouts #) | 21.0 ± 0.6 | 24.2 ± 2.5 | *U*=12.000; p=0.390 |
| | | | |
| **DD** | | | |
| Period (tau; hr) | 23.6 ± 0.1 | 23.4 ± 0.2 | *t*(10)=0.919; p=0.380 |
| Rhythmic power (% variation) | 47.4 ± 1.2 | **39.1 ± 2.0*** | **t(10)=5.319; p=0.003** |
| Cage activity (rev) | 27,225 ± 1495 | 27914 ± 3,218 | *t*(10)=−0.213; p=0.836 |
| Onset variability (min) | 16.3 ± 2.9 | **42.5 ± 7.9*** | **t(10)=−3.387; p=0.007** |
| Fragmentation (bouts #) | 21.3 ± 1.2 | 24.1±1.8 | *t*(10)=−1.416; p=0.187 |

along with decreased sociability. The possibility of impaired social recognition was further tested with the five-trial social test (*Figure 6C*). In this test, the first stranger mouse becomes the familiar mouse after four exposures to the testing mouse. When the second mouse is introduced in the fifth trial, the testing mouse typically shows a boosted interest in investigating the novel mouse. As expected, when the second stranger mouse was introduced to the testing animals in the fifth trial, the WT showed elevated social behavior, while the mutants did not display increased interest in exploring the second novel mouse (*Figure 6C* and *Table 5*). No genotypic differences in social interaction in the first four trials were observed, indicating that habituation was unaltered in the mutants (*Figure 6—figure supplement 1*).

Next, when the active-phase repetitive behavior was examined using the marble bury test, the *Fmr1* KO spent more time digging and buried more marbles compared to WT (*Figure 6D, E* and *Table 5*). When the 30-min trial was divided into three 10-min intervals, the repetitive digging behavior was significantly higher in the *Fmr1* mutants compared with the WT in all three intervals. Two-way ANOVA demonstrated that there was a significant effect of genotype ($F_{(1, 107)}$ = 11.04; p = 0.001) but not of the interval ($F_{(2, 107)}$ = 0.42; p = 0.66). The *Fmr1* KO mice also exhibited significantly more grooming behavior than their WT counterpart (*Figure 6F* and *Table 5*).

Finally, we used these data to determine if disrupted sleep correlated with autistic-like behavioral deficits. Sleep duration (*Figure 6G, H* and *Table 6*) exhibited a moderate to strong correlation with both social recognition and grooming time, while sleep fragmentation (measured by sleep bouts number) only correlated with the latter (*Figure 6J*); the length of sleep bouts (*Table 6*) showed moderate correlation with both social recognition and repetitive behavior. In addition, a moderate correlation was seen between grooming time and the circadian parameters, rhythmic power, and activity onset variability (*Table 6*). In short, our work suggests that even when tested during their circadian active phase, the *Fmr1* KO mice exhibit robust repetitive and social behavioral deficits. Moreover, the shorter and more fragmented the daytime sleep, the more severe the behavioral impairment in the mutants.

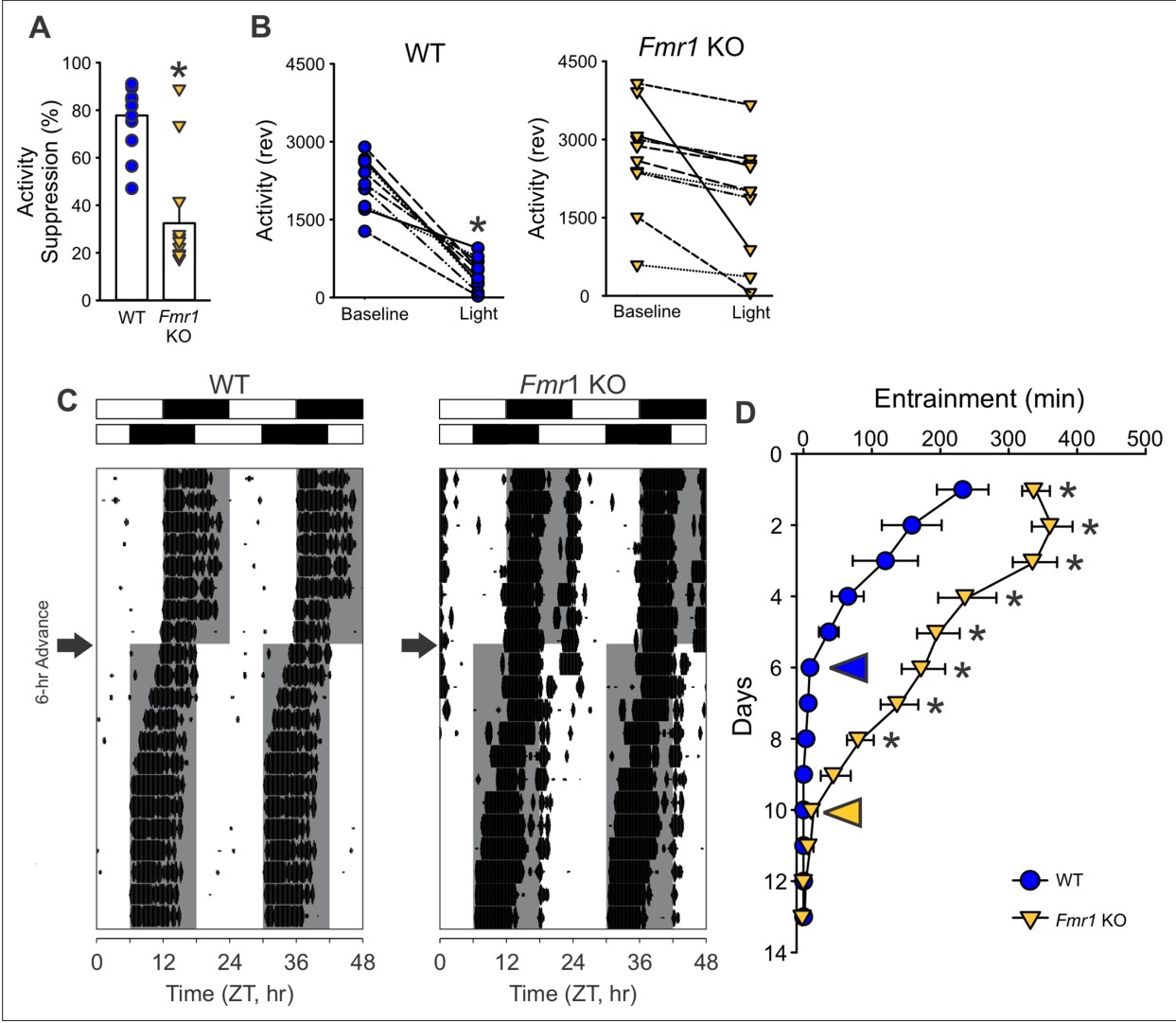

**Figure 3.** The *Fmr1* KO mice display deficits in light-regulated circadian behaviors. (**A, B**) Photic-suppression (masking) of activity in response to a 1 hr light (300 lx, 4500 K) pulse at ZT 14 (lights off; *n* = 10/genotype). The activity level during the light exposure was compared to the activity level during the equivalent hour (ZT 14–15) on the day before the treatment (baseline activity). (**A**) The genotypic difference in the fold change was determined by *t*-test, with the mutants showing a significantly reduced suppression of activity as compared to the WT (*p = 0.05). (**B**) Changes in the activity levels of each individual mouse during the baseline window and the light masking were analyzed using a paired *t*-test. Within-genotype comparisons showed significant suppression in WT (p < 0.001), but not in KO mice (p = 0.12). (**C, D**) Entrainment induced by a 6-hr-phase advanced LD cycle. Representative actograms of light-induced phase shifts of wheel-running activity rhythms (**C**). The white/black bars on the top of actograms indicate the LD cycle before (upper) and after (lower) the 6-hr phase advance. The gray shading in the waveforms indicates the dark phase time--period. The arrows next to the actograms indicate the day when the 6-hr phase advance was applied. (**D**) Quantification of the days to re-entrain shows that the KO mice required more time to adjust (two-way ANOVA: genotype effect $F_{(1,285)}$ = 130.157, p < 0.001; followed by the Holm–Sidak's multiple comparisons test *p < 0.001). The entrainment shifting in the WT (blue circle) and the *Fmr1* KO (yellow triangle) was quantified by the difference between the activity onset and the new ZT12 on each day. The yellow and blue arrowheads in the graph indicate the day when the activity rhythms are considered well entrained. See **Table 3**.

## Amelioration of the behavioral disruptions by scheduled feeding in the *Fmr1* KO mice

The correlations between fragmented sleep and the severity of other behavioral phenotypes support the possibility that interventions focused on improving circadian rhythms and sleep may benefit the observed deficits in the *Fmr1* KO mice. Scheduled feeding can be a powerful regulator of circadian rhythms (***Long and Panda, 2022***; ***Manoogian et al., 2022***) and has been shown to be effective in several disease models (***Whittaker et al., 2018***; ***Wang et al., 2018***; ***Gupta et al., 2022***; ***Whittaker***

**Table 3.** Deficits in circadian light response in the *Fmr1* KO mice.

The circadian light response of male adult WT and *Fmr1* KO mice was evaluated using four behavioral assays and wheel-running activity. <u>First</u>, masking or suppression of activity that occurs when mice are exposed to 1 hr of light during the night at ZT 14 (*n* = 10 per group). <u>Second</u>, the number of days required for the activity rhythms to re-synchronize to a 6-hr phase advance of the LD cycle (*n* = 11 per group). <u>Third</u>, the mice were held on a skeleton photoperiod (1:11:1:11 h LD) and basic locomotor activity parameters were measured. <u>Fourth</u>, to measure the magnitude of a light-evoked phase shift of the circadian system, mice were held in constant dark (DD) and exposed to light for 15 min at CT 16 (*n* = 8 per group). Values are shown as the averages ± SEM. If the assumptions of normality and equal variance were met, a *t*-test was used to analyze the data; otherwise, the Mann–Whitney rank sum test was used. Asterisks indicate significant differences between genotypes. Alpha = 0.05. Degrees of freedom are reported between parentheses. Bold values indicate statistically significant differences.

| | WT | Fmr1 KO | Stats |
|---|---|---|---|
| Masking (% suppression) | 77.7 ± 5.8 | **32.5 ± 10.0*** | $t(18)=4.131; p=0.006$ |
| Re-entrainment (days) | 5.9 ± 0.6 | **11.3 ± 0.4*** | $t(20)=-6.868; p<0.001$ |
| Skeleton photo period (SPP) | | | |
| Period (tau; h) | 24.0 ± 0.0 | **23.7 ± 0.2*** | $U=10.500; p=0.021$ |
| Rhythmic power (% variation) | 39.4 ± 2.0 | **24.0 ± 1.6*** | $t(14)=6.112; p<0.001$ |
| Cage activity (rev) | 21,711 ± 1662 | 25,257 ± 2954 | $t(14)=-1.119; p=0.282$ |
| Activity in the day (%) | 5.29 ± 1.5 | **34.7 ± 6.0*** | $U=0.000; p<0.001$ |
| Onset variability (min) | 18.1 ± 2.8 | **59.5 ± 9.1*** | $t(14)=-4.668; p<0.001$ |
| Light-evoked phase shift (min) | −135.6 ± 26.9 | −64.0 ± 7.7* | $U=5.500; p=0.011$ |
| **Skeleton photo period (SPP)** | | | |
| Period (tau; hr) | 24.0 ± 0.0 | **23.7 ± 0.2*** | $U=10.500; p=0.021$ |
| Rhythmic power (% variation) | 39.4 ± 2.0 | **24.0 ± 1.6*** | $t_{(14)}=6.112; p<0.001$ |
| Cage activity (rev) | 21,711 ± 1662 | 25,257 ± 2954 | $t_{(14)}=-1.119; p=0.282$ |
| Activity in the day (%) | 5.29 ± 1.5 | **34.7 ± 6.0*** | $U=0.000; p<0.001$ |
| Onset variability (min) | 18.1 ± 2.8 | **59.5 ± 9.1*** | $t_{(14)}=-4.668; p<0.001$ |
| Light-evoked phase shift (min) | −135.6 ± 26.9 | **−64.0 ± 7.7*** | $U=5.500; p=0.011$ |

et al., 2023). To test whether circadian reinforcement could ameliorate the observed deficits in the mutants, a new cohort of WT and *Fmr1* KO mice was subjected to a time-restricted feeding (TRF) protocol (6 hr feeding/18 hr fasting; ***Figure 2—figure supplement 1***) for 2 weeks and compared to counterpart animals held on ad libitum feeding (ALF). Both genotypes were able to well adapt to the TRF regimen and ate as much as their ALF counterpart after 5 days; at the end of the 2 weeks, the TRF groups weighed less than the ALF groups (***Figure 7—figure supplement 1***).

The TRF treatment benefited several aspects of the temporal patterning of activity and sleep (***Figure 7***) in both genotypes. The scheduled feeding improved the power of the rhythms and markedly restored the cycle-to-cycle variability in the activity onset of the mutants to WT levels (***Figure 7C–E***), while reducing fragmentation (less activity bouts) in both WT and KO (***Table 7***). Total cage activity was not altered by the intervention in either genotype. By the second week, TRF also improved sleep behavior in both genotypes (***Figure 7F, G***), with the *Fmr1* KO mice showing increased total daytime sleep (***Figure 7H*** and ***Table 7***), reduced sleep fragmentation, i.e. fewer bouts (***Figure 7I***) of longer bout durations (***Figure 7J***) as compared to their ALF counterpart. In general, strong genotypic

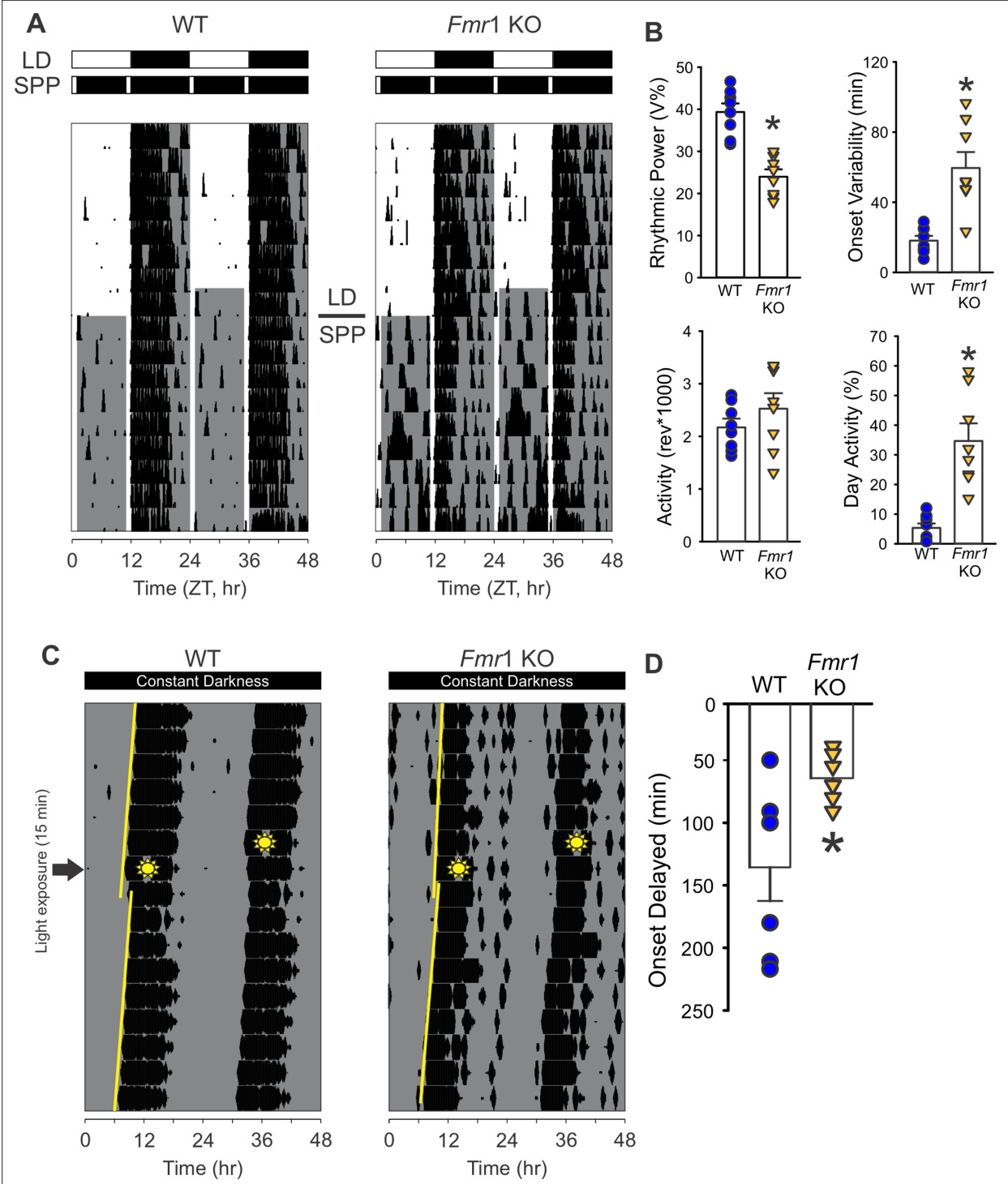

**Figure 4.** The *Fmr1* KO mice exhibit difficulty in adapting to the skeleton photic period (SPP). (**A**) Representative actograms of daily rhythms in cage activity under standard LD cycles (2 weeks) followed by the SPP (1 hr light:11 hr dark:1 hr light:11 hr dark) in WT (left) and *Fmr1* KO (right) mice. The white/black bars on the top of actograms indicate the baseline LD cycle (upper) and the SPP LD cycles (lower). The gray shading in the waveforms indicates the time of the dark phases. (**B**) Measures of locomotor activity rhythms under SPP. Many of the parameters measured were significantly different between the genotypes, with the mutants being more impacted and showing lower rhythmic power and increased variability of the activity onset. In addition, the mutants displayed higher activity during their day. Histograms show the means ± SEM with the values from each individual animal overlaid. Significant differences (p < 0.05), determined by *t*-test or Mann–Whitney test, are indicated with an asterisk. (**C, D**) Light-induced phase delay of free-running activity rhythms in mice exposed to light (300 lx, 4500 K) for 15 min at circadian time (CT) 16. Mice were held in constant darkness. By definition, CT 12 is the beginning of the activity cycle in DD for a nocturnal organism. Examples of light-induced phase shifts of wheel-running activity

*Figure 4 continued on next page*

*Figure 4 continued*

rhythms (**C**) of WT (left) and *Fmr1* KO (right) and quantified phase delay (**D**). In the representative actograms, the yellow lines indicate the best-fit line of the activity onset across the 10 days before and after the light pulse. The amount of phase delay is determined by the difference between the two lines on the day after the light pulse. The sunny-shape symbols indicate when the mice were exposed to light (CT16). Compared to WT, the *Fmr1* KO showed reduced phase shift of their activity rhythms (Mann–Whitney *U*, *p = 0.011). See *Table 3*.

differences were present in the response to TRF with the WT mice showing a more robust sleep response (*Table 7*).

The benefits of TRF were also evident on other aberrant behaviors in the *Fmr1* KO model, and after 2 weeks of treatment, this feeding regimen improved social memory recognition while reducing the inappropriate grooming behavior (*Table 8*). Scheduled feeding improved social memory in the KO mice, by restoring the preference for the novel over the familiar mice to WT-like level (*Figure 8A* and *Table 8*), suggesting that, perhaps, the treated mutants were able to distinguish the novel mouse from the familiar mouse. Grooming behavior was also significantly reduced by TRF in the *Fmr1* KO mice, which showed a significant reduction in the time spent self-grooming (*Figure 8B*), without affecting their locomotor activity (*Figure 8C*). These data suggest that TRF has ameliorating effects on sleep/wake rhythms as well as autistic-like behaviors in the *Fmr1* KO model.

## Attenuation of the elevated levels of pro-inflammatory markers in the *Fmr1* KO mice by TRF

An abnormal immune response has been suggested to play a role in FXS pathophysiology (*Reynolds et al., 2021*; *Dias et al., 2023*; *Robinson-Agramonte et al., 2022*). Hence, blood was collected at the end of the 2 weeks of the scheduled feeding regimen to investigate whether TRF would impact the profile of pro-inflammatory markers in the *Fmr1* KO mice. The mutants on ALF displayed elevated levels of several cytokines in comparison with their WT counterparts (*Table 9*), in particular interleukin-12 (IL-12), interferon-gamma (IFN$_\gamma$), and the chemokine ligand-9 (CXCL-9). These genotypic differences were abolished in the TRF-treated groups (*Figure 9A*). Scheduled feeding also repristinated the *Fmr1* KO levels of IL-2 to WT levels. Furthermore, elevated IL-12 and IFN-γ levels were significantly associated with markers of poor sleep, such as higher activity in the light phase, shorter sleep time/sleep bout length, and higher numbers of sleep bouts (*Table 10*), as well as with social memory deficits and excessive grooming (*Figure 9B, C* and *Table 10*). These data suggest that TRF-dampening of the pro-inflammatory signaling in *Fmr1* KO mice provides a potential mechanism to explain, at least in part, the observed behavioral improvements.

## Discussion

Sleep and circadian rhythm disruptions are prevalent in individuals with NDDs, including FXS. Common issues include difficulty initiating and maintaining sleep and irregular sleep patterns, which significantly impair the quality of life of FXS individuals and their families/caretakers (*Budimirovic et al., 2022*; *Minhas et al., 2025*). Notably, individuals exhibiting more severe behavioral symptoms often experience more pronounced sleep problems (*Schreck et al., 2004*; *Taylor et al., 2012*; *Kaufmann et al., 2024*), suggesting that circadian and sleep dysfunction exacerbate core NDD features. These disturbances are detectable quite early, even in infants and toddlers carrying the FMR1 mutation (*D'Souza et al., 2020*), raising the possibility that disrupted sleep/wake cycles may be an initial and influential feature of disease expression. Mechanistically, such disruptions could stem from direct effects of FMR1 loss on sleep and circadian circuits, or indirectly from secondary symptoms such as anxiety, sensory hypersensitivity, and repetitive behaviors.

Animal models of FXS, such as the *Fmr1* KO mouse, are essential for dissecting the underlying mechanisms and testing interventions (*Sandoval et al., 2024*). In the present study, we demonstrate that loss of *Fmr1* results in disrupted sleep and circadian rhythms, including higher fragmentation of the rest-phase sleep and abnormal diurnal activity patterns. These disruptions were associated with deficits in social memory and increased repetitive behaviors. Importantly, we found that a circadian-based intervention (TRF) not only improved rhythmicity and sleep consolidation but also ameliorated behavioral deficits and reduced peripheral inflammatory markers.

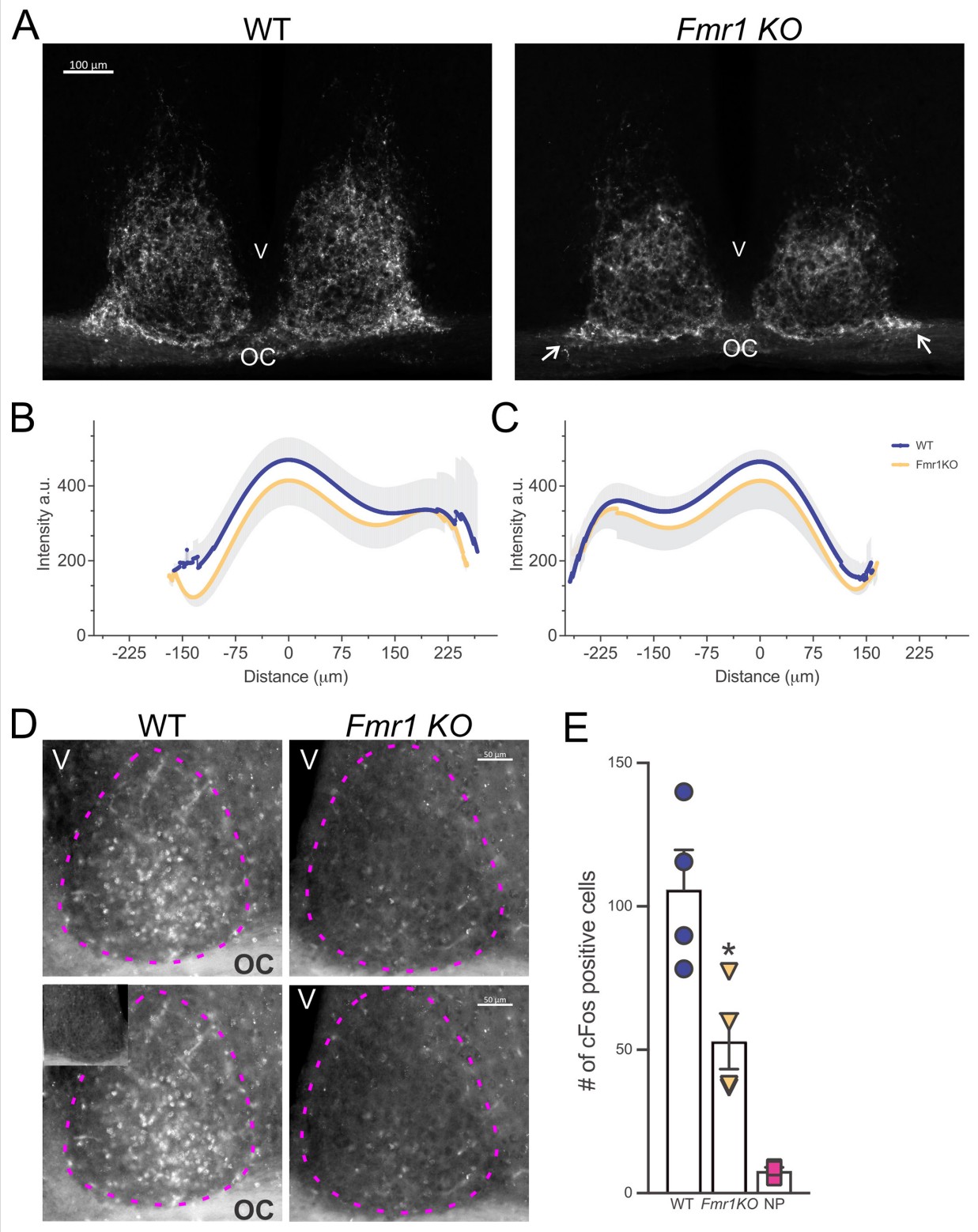

**Figure 5.** Abnormal retinal-suprachiasmatic nucleus (SCN) connectivity in the *Fmr1* KO mice. To trace the projections from the retina to the SCN via the retino-hypothalamic tract (RHT), WT and *Fmr1* KO mice received a bilateral intravitreal injection of Cholera Toxin (β-subunit) conjugated to Alexa Fluor555 and were perfused 72 hr later. (**A**) Lower intensity of the fluorescently labeled RHT projections can be observed both laterally and medially to the ventral part of the SCN in the *Fmr1 KO* mice (white arrows) as compared to WT, suggesting a loss of afferent projections to the SCN. (**B, C**) Densitometric analysis of the distribution of the Cholera Toxin fluorescence intensity in the ventral SCN (see also *Figure 5—figure supplement 1*) of

*Figure 5 continued on next page*

**Table 4.** Compromised retinal–SCN connectivity.

Subtle decrease in the relative intensity of Cholera Toxin (β subunit) in the retinal afferents to the suprachiasmatic nucleus (SCN) of *Fmr1* KO mice. There was a stronger impact of the loss of FMRP on the induction of light-evoked cFos expression in the SCN. Control no pulse = WT mice held in DD but not exposed to the light pulse at CT16. Histomorphometrical analysis of the SCN revealed no differences between WT and *Fmr1* male mice. All measurements were performed by two independent observers masked to the experimental groups. Results are shown as the mean ± SD. Alpha = 0.05. Degrees of freedom are reported between parentheses. Bold values indicate statistically significant differences.

| Cholera Toxin | WT (*n* = 3) | Fmr1 KO (*n*=3) | Mann–Whitney test |
|---|---|---|---|
| Mean intensity (a.u.) | 247.2 ± 52.8 | 221.8 ± 29.5 | *U*=2; p=0.4000 |
| | | | |
| **cFos immunopositive cells (#)** | WT (*n*=4) | *Fmr1* KO (*n*=4) | Control no pulse (*n*=3) |
| | 105.9 ± 27.5 | **52.96 ± 19.4\*** | 7.720 ± 1.28 |
| One-way ANOVA | $F_{(2,8)}$**=19.75; p=0.0008** | | |
| | | | |
| | WT (*n*=6) | Fmr1 KO (*n*=6) | Mann–Whitney test |
| Area (μm²) | 308,184 ± 6198 | 294,721 ± 11,561 | *U*=11; p=0.3095 |
| Perimeter (μm) | 1178 ± 40.84 | 1154 ± 60.82 | *U*=11; p=0.3095 |
| Height (μm) | 395.2 ± 21.69 | 386.9 ± 20.13 | *U*=11; p=0.3095 |
| Width (μm) | 338.1 ± 16.05 | 333.7 ± 24.08 | *U*=12; p=0.3939 |

## Circadian and sleep disturbances in *Fmr1* KO mice

Our behavioral sleep assay revealed reduced sleep duration and increased fragmentation during the light/rest phase in *Fmr1* KO mice (**Figure 1**), consistent with a previous report by *Saré et al., 2017*. In contrast, *Westmark et al., 2023*, using EEG, did not observe baseline sleep differences between genotypes, possibly reflecting methodological differences or subtle phenotypic variability. A key gap in the field is the lack of studies probing sleep homeostasis in *Fmr1* KO mice; future studies using sleep deprivation protocols may reveal underlying vulnerabilities. Notably, the *Fmr1* KO mice exhibit neural oscillation abnormalities, including increased resting gamma power and enhanced auditory-evoked responses, paralleling the EEG signatures in FXS patients (*Jonak et al., 2024*).

Assessment of circadian locomotor activity revealed weakened rhythmicity, elevated daytime activity, and a greater onset variability in *Fmr1* KO mice (**Figure 2**), consistent with prior studies (*Bonasera et al., 2017*; *Angelakos et al., 2019*). These deficits persisted in constant darkness (DD), indicating a compromised endogenous circadian clock. Differences in activity rhythms measured in LD conditions could be driven by the circadian clock or the direct effects of light and dark. Although earlier studies using Fmr1/Fxr2 double mutants reported severe rhythm loss (*Zhang et al., 2008*), the present study demonstrates that the loss of *Fmr1* alone is sufficient to impair circadian regulation.

*Figure 5 continued*

WT and *Fmr1* KO mice. The intensity peaks of the profile plot of four to five consecutive coronal sections containing the middle SCN were aligned and then averaged to obtain a single curve per animal. Results are shown as the mean ± standard deviation (SD) for the left (**B**) and the right (**C**) SCN of each genotype. (**D, E**) Light-induction of cFos was greatly reduced in the SCN of the *Fmr1* KO mice compared to WT. Mice held in DD were exposed to light (300 lx, 4500 K) for 15 min at CT 16 and perfused 45 min later (CT 17). (**D**) Representative serial images of light-evoked cFos expression in the SCN. The inset in the lower left panel shows the lack of cFos immunopositive cells in the SCN of mice held in DD but not exposed to light. Dotted magenta lines delineate the SCN. OC = optic chiasm, V = third ventricle. (**E**) The number of immune-positive cells in the left and right SCN from 3 to 5 consecutive coronal sections per animal were averaged to obtain one number per animal and are presented as the mean ± SD per genotype. One-way ANOVA followed by Bonferroni's multiple comparisons test, \*p = 0.0201. See **Table 4**.

The online version of this article includes the following figure supplement(s) for figure 5:

**Figure supplement 1.** Histomorphological Analysis and SCN Landmarks.

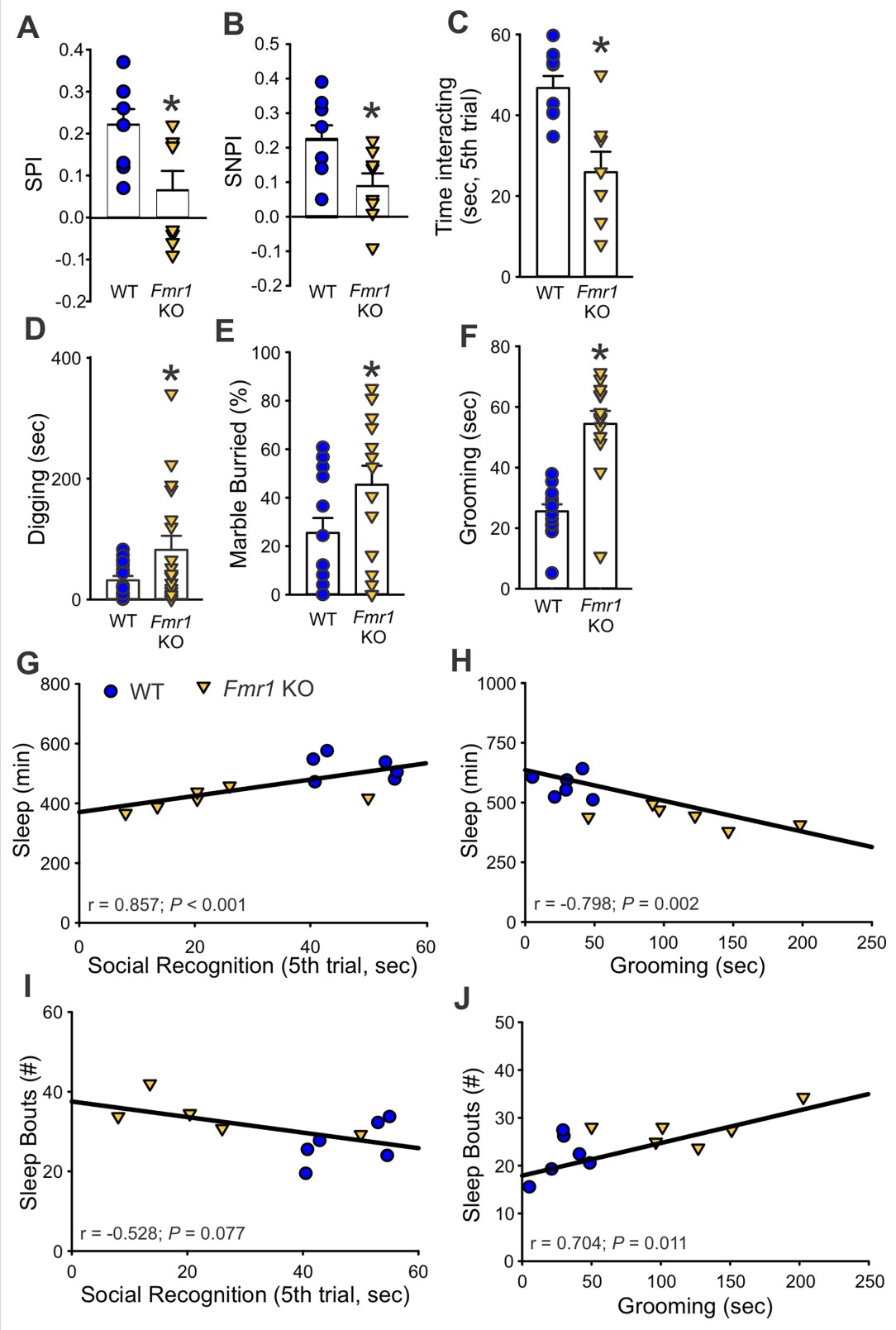

**Figure 6.** The deficits in social recognition and repetitive behaviors of the *Fmr1* KO mice correlate with altered sleep behavior. <u>Tests of social behavior</u>: (**A**) In the first stage of the three-chamber social test, when the testing mouse is given a choice between a stranger mouse and an inanimate object, the *Fmr1* KO mice spent less time with the stranger mouse and had a lower social preference index (SPI) than the WT. (**B**) In the second stage, the testing mouse is given the choice between a chamber with a novel mouse and the one with the familiar mouse. The mutants spent less time with the

*Figure 6 continued on next page*

**Table 5.** *Fmr1* KO mice present with deficits in social discrimination.

Comparisons of social discrimination behavior in age-matched WT and *Fmr1* KO male mice (*n* = 8 per group) were assessed using the three-chamber and the five-trial social interaction test. Social preference index (SPI) = difference in the time spent with the novel mouse and object divided by the sum of the time spent with the novel mouse and the object. Social novelty preference index (SNPI) = difference in the time spent with the novel and familiar mouse divided by the sum of the time spent with both the novel and familiar mice. The repetitive behavior in WT and *Fmr1* KO mice was (*n* = 14/genotype) assessed using the marble bury and grooming tests. Values are shown as the averages ± SEM. If the assumptions of normality and equal variance were met, a *t*-test was used to analyze the data; otherwise, the Mann–Whitney test was used. Alpha = 0.05. Degrees of freedom are reported between parentheses. Bold values indicate statistically significant differences.

| | WT | *Fmr1* KO | Statistical values |
|---|---|---|---|
| **Three-chamber social test** | | | |
| Time with object (s) | 154 ± 7.4 | 184 ± 13.3 | $t_{(14)}$=−2.100; p=0.054 |
| Time with mouse (s) | 236 ± 13.6 | 210 ± 14.5 | $t_{(14)}$=1.777; p=0.097 |
| SPI | 0.20 ± 0.04 | **0.06 ± 0.05\*** | ***U*=12.500; p=0.038** |
| **Three-chamber social recognition test** | | | |
| Familiar-mouse chamber (s) | 143 ± 11.2 | **173 ± 1.7\*** | ***U*=11.000; p=0.028** |
| Novel-mouse chamber (s) | 225 ± 13.5 | 209.5 ± 14.2 | $t_{(14)}$=0.879; p=0.394 |
| SNPI | 0.22 ± 0.04 | **0.09 ± 0.04\*** | **$t_{(14)}$=2.445; p=0.028** |
| **Five-trial social recognition test** | | | |
| Familiar mouse (trials 1–4, s) | 154 ± 15.3 | 150 ± 20.9 | $t_{(14)}$=0.153; p=0.881 |
| Novel mouse (trial 5, s) | 46.7 ± 2.8 | **25.9 ± 4.7\*** | **$t_{(14)}$=3.781; p=0.002** |
| **Marble bury test** | | | |
| Digging in total of 30 min (s) | 19.8 ± 4.2 | **106 ± 28.0\*** | ***U*=30.000; p=0.002** |
| Buried marbles (%) | 19.6 ± 6.1 | **45.8 ± 8.6\*** | **$t_{(26)}$=−2.255; p=0.017** |
| Distance traveled (m) | 49.1 ± 3.5 | **63.9 ± 2.7\*** | **$t_{(26)}$=−3.497; p=0.002** |
| **Grooming test** | | | |
| Grooming (s) | 25.5 ± 2.3 | **54.4 ± 4.3\*** | ***U*=13.000; p<0.001** |
| Distance traveled (m) | 67.7 ± 2.6 | **78.0 ± 2.4\*** | **$t_{(26)}$=−2.985; p=0.006** |

*Figure 6 continued*

novel mouse compared to the familiar one, and also in this phase exhibited lower social novelty preference (SNPI) as compared to the WT. (**C**) The possibility of reduced social recognition was further tested with the five-trial social test. In this test, the stranger mouse becomes a familiar mouse after four exposures to the testing mouse, then a novel mouse is introduced in the fifth trial. The WT mice showed a higher interest in the novel mouse compared to the *Fmr1* KO mice. Tests of repetitive behaviors: The amount of digging in the bedding (**D**) and the percentage of marbles buried (**E**) were measured with the marble bury test. The *Fmr1* KO mice spent longer time digging and buried more marbles compared to WT. (**F**) Grooming behavior, assessed in a novel arena, was significantly higher in the *Fmr1* KO mice as compared to WT. Histograms show the means ± SEM with the values from the individual animals overlaid. Significant differences (\*p < 0.05) were determined by *t*-test or Mann–Whitney test. See also *Table 5*. Sleep duration (min, **G**, **H**) correlated with impaired social recognition and abnormal grooming behaviors. Sleep fragmentation, measured by number of sleep bouts, correlated only with grooming (**I, J**) (Pearson correlation test). See *Table 6*.

The online version of this article includes the following figure supplement(s) for figure 6:

**Figure supplement 1.** To assess social recognition memory, Fmr1 KO and WT mice underwent a five-trial social interaction paradigm in a neutral open-field arena.

**Table 6.** Correlation between sleep disturbances and the severity of impaired behaviors in the *Fmr1*KO mice.
Data obtained from age-matched WT and *Fmr1* KO mice housed under standard LD cycles were tested for associations with the Pearson correlation test. The most prominent sleep phenotypes were usually observed during the animals' light-phase sleep; hence, only measures between ZT 0 and 12 were used for these analyses. The correlation coefficients are reported; those significant are shown in bold and labeled with an asterisk. Alpha = 0.05.

| | Three-chamber social test (SNPI) | Social recognition (trial 5, s) | Digging (s) | Marble buried (%) | Grooming (s) |
|---|---|---|---|---|---|
| Rhythmic power (% V) | 0.18 | 0.43 | 0.13 | –0.051 | **–0.62*** |
| Onset variability (min) | –0.37 | –0.4 | 0.23 | **0.61*** | **0.63*** |
| Sleep duration (min) | 0.47 | **0.86*** | –0.3 | –0.52 | **–0.8*** |
| Sleep bout (#) | –0.39 | –0.53 | 0.27 | **0.78*** | **0.7*** |
| Avg. sleep bout length (min) | 0.44 | **0.59*** | –0.29 | **–0.71*** | **–0.75*** |
| MAX sleep bout length (min) | 0.49 | **0.76*** | –0.4 | **–0.65*** | **–0.78*** |

## Impaired light response and SCN connectivity in the *Fmr1* KO mice

To explore the basis of the observed dysregulations, we evaluated the photic input to the circadian system. *Fmr1* KO mice displayed impaired negative masking, delayed re-entrainment to phase shifts, blunted light-induced phase shifts, and poor entrainment to a skeleton photoperiod (*Figures 3 and 4*). These behavioral deficits suggest a compromised signal transmission from the retina to the SCN via the ipRGCs. Such a hypothesis was confirmed by anatomical and functional assays; indeed, the ipRGC innervation of the SCN was lessened in the *Fmr1* KO mice (*Figure 5A*), and the light-evoked c-Fos induction was substantially reduced (*Figure 5B*). Given that the FMRP is expressed in the ipRGCs (*Zhang et al., 2020*), its loss may underline the deficient light response observed in the mutants. Interestingly, this reduced sensitivity to circadian light input contrasts with the general sensory hypersensitivity typical of FXS, highlighting a unique dissociation in sensory processing pathways (*Rais et al., 2018*). Whether these circadian-specific photic impairments are present in FXS patients warrants further investigation.

Despite these light input abnormalities, the *Fmr1* KO mice did not exhibit a shortened free-running period (tau) in DD (*Figure 2D*; *Angelakos et al., 2019*), nor were the rhythmic expressions of the clock genes altered (*Zhang et al., 2008*). These findings suggest that the core molecular clock remains intact, and the circadian dysfunction may arise from an altered excitatory/inhibitory (E/I) balance or impaired network synchronization within the SCN (*Olde Engberink et al., 2023*). Hence, our present results reinforce that loss of *Fmr1* impairs the circadian system at the level of input and output regulation rather than at the core molecular oscillation.

## Sleep and behavior correlate in the *Fmr1* KO mice

The *Fmr1* KO model reproduces core behavioral deficits of FXS, including impaired social recognition and increased repetitive behaviors (*Melancia and Trezza, 2018*; *Kat et al., 2022*). By analyzing sleep, circadian activity, and behavior in the same animals, we identified significant correlations: mice with more disrupted sleep and weaker circadian rhythms exhibited more severe social and repetitive behavior impairments (*Figure 6*). These findings parallel clinical studies showing that disrupted sleep predicts next-day behavioral difficulties in FXS and autism spectrum disorders (*Cohen et al., 2018*; *Robinson-Shelton and Malow, 2016*; *Schwichtenberg et al., 2022*; *Minhas et al., 2025*). Although correlational, these results suggest that targeting circadian and sleep dysfunction may offer a novel route for ameliorating some behavioral phenotypes present in certain NDDs.

## TRF as a therapeutic strategy

Given the link between circadian disruption and behavior, we tested whether restoring rhythmicity could improve outcomes. TRF, a circadian-aligned feeding schedule, significantly improved the power of the rhythms, reduced cycle-to-cycle (onset) variability, and consolidated daytime sleep (*Figure 7*). Importantly, TRF reduced repetitive behavior and improved social memory and interactions in the

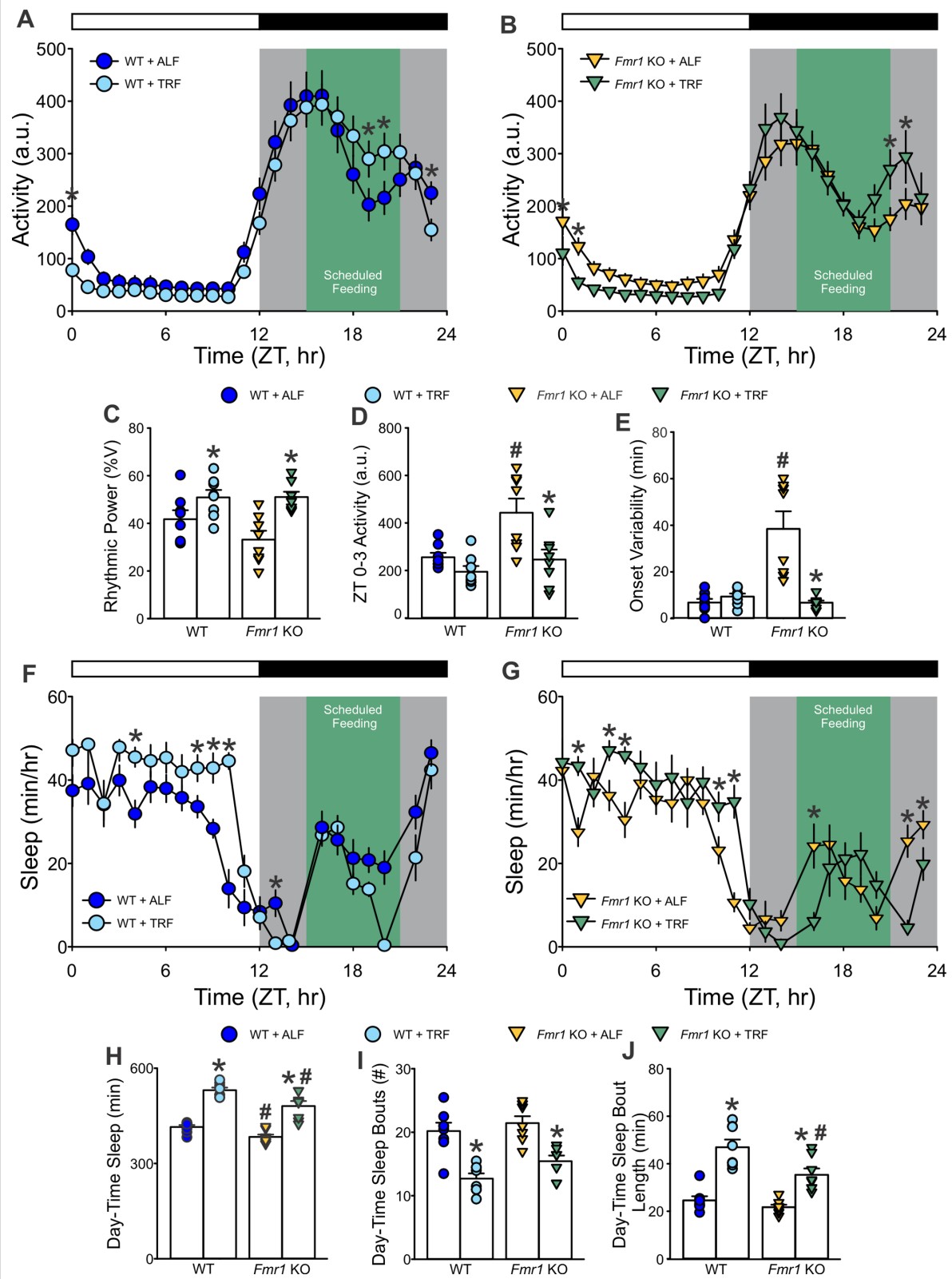

**Figure 7.** Amelioration of sleep/wake rhythms in the *Fmr1* KO mutants by time-restricted feeding (TRF). (**A, B**) Waveforms of daily rhythms in cage activity using infrared (IR) detection in the WT (circle) and *Fmr1* KO (triangle) mice under ad libitum feeding (ALF) or TRF. The activity waveforms (1 hr bins) were analyzed using a three-way ANOVA with genotype, feeding regimen, and time as factors followed by Holm–Sidak's multiple comparisons test. There were significant effects of genotype ($F_{(1, 767)}$ = 13.301; $p < 0.001$) and time ($F_{(23, 767)}$ = 94.188; $p < 0.001$), as well as significant interactions

*Figure 7 continued on next page*

**Table 7.** Scheduled feeding improved sleep/wake rhythms in the *Fmr1* KO mutants.

Locomotor activity rhythms and immobility-defined sleep were recorded from WT and *Fmr1* KO mice on ad libitum feeding (ALF) or time-restricted feeding (TRF; n = 8 per group). As the running wheels interfere with the feeders, we used infrared (IR) to measure the activity rhythms in these experiments. Since the most prominent sleep phenotypes were observed during the light-phase sleep and sleep recordings were paused during the dark phase for adding (ZT15) and removing (ZT21) food, the analyses below only focused on the effects of TRF on sleep during the light-phase sleep (ZT 0–12). Values are shown as the averages ± SEM. Data were analyzed by two-way ANOVA with genotype and treatment as factors, followed by the Holm–Sidak's multiple comparisons test. Asterisks indicate significant differences within genotype – different feeding regimen, while crosshatch indicates significant differences between genotypes – same feeding regimen. Alpha = 0.05. Degrees of freedom are reported between parentheses. Bold values indicate statistically significant differences.

| | WT | | *Fmr1* KO | | Two-way ANOVA | | |
|---|---|---|---|---|---|---|---|
| Activity rhythms | ALF | TRF | ALF | TRF | Genotype | Treatment | Interaction |
| Rhythmic power (% variation) | 41.8 ± 3.8 | **50.9 ± 3.1\*** | 33.2 ± 3.8 | **51.1 ± 2.2\*** | $F_{(1, 31)}$=1.9; p=0.18 | **$F_{(1, 31)}$=19.41; p<0.001** | $F_{(1, 31)}$=2.04; p=0.17 |
| Cage activity (a.u.) | 4307 ± 517 | 4064 ± 396 | 3748 ± 402 | 3750 ± 471 | $F_{(1, 31)}$=1.08; p=0.31 | $F_{(1, 31)}$=0.082; p=0.78 | $F_{(1, 31)}$=0.085; p=0.77 |
| Light-phase activity (ZT 0–3, a.u.) | 255 ± 18.9 | 194 ± 24.3 | 443 ± 59.6# | **246 ± 42.0\*** | **$F_{(1, 31)}$=10.44; p=0.03** | **$F_{(1, 31)}$=12.14; p=0.002** | $F_{(1, 31)}$=3.38; p=0.07 |
| Onset variability (min) | 6.8 ± 1.6 | 9.4 ± 1.3 | 38.5 ± 7.5# | **6.8 ± 0.9\*** | **$F_{(1, 31)}$=15.58; p<0.001** | **$F_{(1, 31)}$=15.70; p<0.001** | **$F_{(1, 31)}$=21.68; p<0.001** |
| Fragmentation (bouts #) | 29.3 ± 1.7 | **23.4 ± 1.0\*** | 29.5 ± 1.8 | **23.4 ± 1.1\*** | $F_{(1, 31)}$=0.11; p=0.74 | **$F_{(1, 31)}$=21.91; p<0.001** | $F_{(1, 31)}$=0.23; p=0.64 |
| Sleep | | | | | | | |
| Sleep duration (min) | 414 ± 6.3 | **525 ± 11.9\*** | 384 ± 7.4# | **481 ± 15.7\*#** | **$F_{(1, 31)}$=18.35; p<0.001** | **$F_{(1, 31)}$=127.19; p<0.001** | $F_{(1, 31)}$=0.96; p=0.34 |
| Sleep bouts (#) | 20.2 ± 1.3 | **12.7 ± 0.8\*** | 21.4 ± 1.1 | **15.4 ± 0.9\*** | **$F_{(1, 31)}$=4.21; p=0.05** | **$F_{(1, 31)}$=47.94; p<0.001** | $F_{(1, 31)}$=0.59; p=0.45 |
| Avg. sleep bout length (min) | 24.6 ± 1.7 | **47.0 ± 3.2\*** | 21.7 ± 1.04 | **35.3 ± 2.7\*#** | **$F_{(1, 31)}$=11.04; p=0.002** | **$F_{(1, 31)}$=68.3; p<0.001** | $F_{(1, 31)}$=4.03; p=0.055 |
| MAX sleep bout length (min) | 96.4 ± 4.3 | **137 ± 3.3\*** | 88.3 ± 3.1 | **109 ± 4.7\*#** | **$F_{(1, 31)}$=25.03; p<0.001** | **$F_{(1, 31)}$=68.76; p<0.001** | **$F_{(1, 31)}$=7.64; p=0.01** |

*Figure 7 continued*

between genotype and time (p < 0.001) and feeding regimen and time (p < 0.001) on the locomotor activity rhythms of both WT and *Fmr1* KO mice. The green area indicates the time period when the food hoppers were opened for 6 hr between ZT 15 and ZT 21. (**C–E**) Measures of locomotor activity rhythms: Both genotypes exhibited an increase in the power of the rhythms under TRF compared to ALF controls. The increase in early-day and late-night activity as well as the onset variability seen in the *Fmr1* KO mice was corrected by the TRF. Data are shown as the means ± SEM; two-way ANOVA followed by Holm–Sidak's multiple comparisons test with genotype and feeding regimen as factors, *p < 0.05 significant differences within genotypes (different feeding regimen); #p < 0.05 significant differences between genotypes (same feeding regimen). (**F, G**) Waveforms of daily rhythms of immobility-defined sleep. The sleep waveforms (1 hr bins) were analyzed by two-way ANOVA with time and feeding regimen as factors followed by the Holm–Sidak's multiple comparisons test. There were significant effects of time for both WT ($F_{(23, 351)}$ = 9.828, p < 0.001) and *Fmr1* KO ($F_{(23, 351)}$ = 1.806, p = 0.014) mice, but not of feeding regimens. Missing data points precluded the use of three-way ANOVA for these measures. (**H–J**) Measures of immobility-defined sleep in the light phase. Both genotypes held on TRF exhibited an increase in sleep duration and in sleep bout length as well as a reduction in sleep fragmentation, measured by the number of sleep bouts, compared to their ALF counterparts. Data are shown as the means ± SEM; two-way ANOVA followed by Holm–Sidak's multiple comparisons test with genotype and diet as factors, *p < 0.05 significant differences within genotype – between diet regimens; #p < 0.05 significant differences between genotypes – same feeding regimen. See **Table 7**.

The online version of this article includes the following figure supplement(s) for figure 7:

**Figure supplement 1.** Both genotypes well adapt to the feeding regimen.

**Table 8.** Scheduled feeding improved social recognition memory and reduced grooming behavior in the *Fmr1* KO mice.
Adult male WT and *Fmr1* KO mice on ad libitum feeding (ALF) or time-restricted feeding (TRF) (*n* = 8 per group) were exposed to the five-trial social test and the grooming test. Data are shown as the averages ± SEM and were analyzed by two-way ANOVA with genotype and treatment as factors followed by the Holm–Sidak's multiple comparisons test. Asterisks indicate significant differences within genotype – different feeding regimen, while crosshatch those between genotypes – same feeding regimen. Alpha = 0.05. Degrees of freedom are reported between parentheses. Bold values indicate statistically significant differences.

| | WT | | *Fmr1* KO | | Two-way ANOVA | | |
|---|---|---|---|---|---|---|---|
| Measures | ALF | TRF | ALF | TRF | Genotype | Treatment | Interaction |
| Familiar mouse (trials 1–4, s) | 80.5 ± 12 | 96.5 ± 14 | 67.5 ± 4.7 | 95.8 ± 15 | $F_{(1, 31)}=0.36$; p=0.55 | $F_{(1, 31)}=3.77$; p=0.062 | $F_{(1, 31)}=0.29$; p=0.59 |
| Novel mouse (trial 5, s) | 43.8 ± 8.9 | 66.1 ± 8.8 | 23.3 ± 4.0# | 50.3 ± 10* | $F_{(1, 31)}=5.37$; **p=0.028** | $F_{(1, 31)}=9.88$; **p=0.004** | $F_{(1, 31)}=0.092$; p=0.76 |
| Grooming (s) | 19.9 ± 1.2 | 16.9 ± 2.7 | **41.4 ± 1.7#** | **26.2 ± 3.1*#** | $F_{(1, 31)}=60.04$; **p<0.001** | $F_{(1, 31)}=20.74$; **p<0.001** | $F_{(1, 31)}=9.4$; **p=0.005** |
| Distance traveled (m) | 72.2 ± 4.2 | 66.6 ± 4.7 | 67.0 ± 6.6 | 63.2 ± 2.2 | $F_{(1, 31)}=0.98$; p=0.33 | $F_{(1, 31)}=1.14$; p=0.3 | $F_{(1, 31)}=0.039$; p=0.85 |

*Fmr1* KO mice (*Figure 8*), establishing proof-of-principle that circadian-based behavioral interventions can ameliorate core phenotypes of FXS.

Ketogenic diets have previously been shown to improve both sleep and seizure control in the *Fmr1* KO mice, albeit with limited effects on other behaviors (*Westmark et al., 2020*; *Westmark et al., 2023*; *Westmark et al., 2024*). TRF, by contrast, appears to directly impact both behavioral and physiological endpoints, potentially through its entraining effects on peripheral and central clocks. Prior studies have shown that TRF improves behavior in aging and animal models of neurodegenerative disorders (*Wang et al., 2018*; *Whittaker et al., 2018*; *Whittaker et al., 2023*), but this is the first report to demonstrate benefits in a model of NDDs.

## TRF reduces inflammatory signatures in *Fmr1* KO mice

An emerging literature suggests that immune dysregulation accompanies FXS pathology (*Ashwood et al., 2010*; *Careaga et al., 2014*; *Van Dijck et al., 2020*). In the present study, TRF reduced the abnormal levels of IL-12 and IFN-γ in the *Fmr1* KO mice (*Figure 9A* and *Table 9*), and the increased levels of these cytokines correlated with both sleep disturbances and behavioral impairments (*Figure 9B, C* and *Table 10*). While most cytokines did not differ between genotypes, the normalization of IL-12 and IFN-γ levels following TRF supports a mechanistic link between inflammation and behavioral symptoms in FXS. These results are consistent with prior evidence implicating IL-12 in social and repetitive behavior in autism spectrum disorders (*Ashwood et al., 2011*; *Fallah et al., 2020*) and suggest that circadian-based interventions may exert part of their benefit by dampening low-grade inflammation.

Whether TRF exerts direct anti-inflammatory effects in the central nervous system remains unknown. Future studies should assess microglia activation, blood–brain barrier integrity, and central cytokine levels following TRF in the *Fmr1* KO and related models. Moreover, defining the causal role of specific cytokines such as IL-12 and IFN-γ in behavioral regulation may offer novel therapeutic targets for FXS and related disorders.

## Translational implications

The present findings support the view that circadian disruption is not merely a downstream consequence of disease processes but actively contributes to symptom expression. Hence, the possibility that interventions designed to reinforce circadian rhythms can hold therapeutic value for individuals with FXS and related neurodevelopmental conditions. Given that sleep and circadian dysfunction are detectable early in development and are predictive of more severe clinical phenotypes, circadian-based interventions may be particularly beneficial if applied during periods of heightened neural plasticity. Importantly, TRF represents a relatively low-cost, non-invasive strategy that could be feasibly implemented in real-world settings. Further translational work is needed to evaluate whether the mechanistic links identified here—between circadian misalignment, immune dysregulation, and

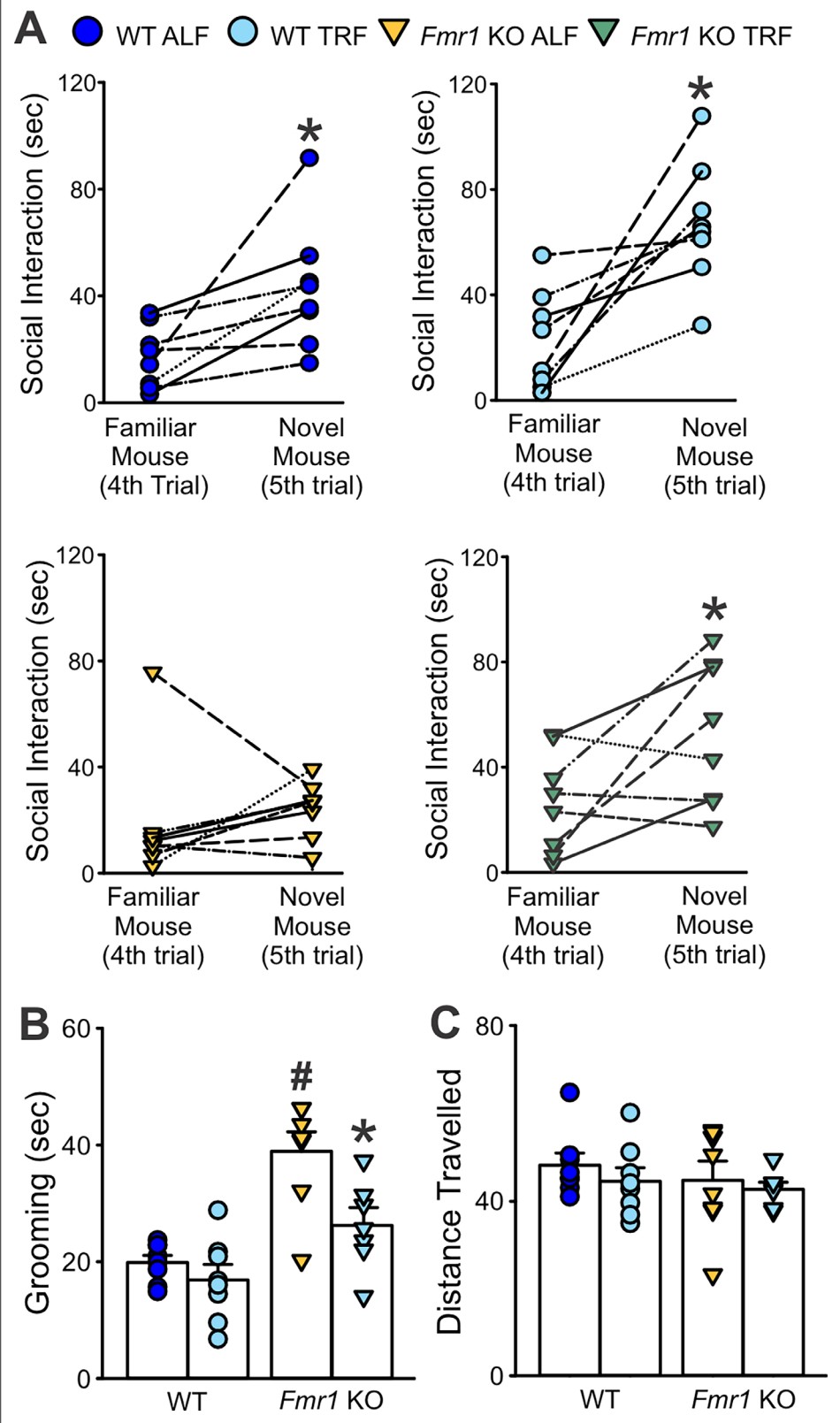

**Figure 8.** Improved social memory and stereotypic grooming behavior in the *Fmr1* KO mice after 2 weeks of time-restricted feeding (TRF). (**A**) Social memory was evaluated with the five-trial social interaction test as described above. The social memory recognition was significantly augmented in the *Fmr1* KO by the intervention, suggesting that the treated mutants were, perhaps, able to distinguish the novel mouse from the familiar mouse. The time

*Figure 8 continued on next page*

**Table 9.** Scheduled feeding affects the levels of plasma cytokines in WT and Fmr1 KO mice.

The levels of several plasma cytokines were measured in WT and mutants held on ad libitum feeding (ALF) or time-restricted feeding (TRF) regimen (n = 8 per group). Values are shown as the averages ± SEM. Data were analyzed by two-way ANOVA with genotype and treatment as factors followed by the Holm–Sidak's multiple comparisons test. Asterisks indicate significant differences within genotype – different feeding regimen, while crosshatch those between genotypes – same feeding regimen. Alpha = 0.05. Degrees of freedom are reported between parentheses. Bold values indicate statistically significant differences.

| Measures | WT | | Fmr1 KO | | Two-way ANOVA | | |
|---|---|---|---|---|---|---|---|
| | ALF | TRF | ALF | TRF | Genotype | Treatment | Interaction |
| TNFα | 4.1 ± 1.0 | 2.6 ± 0.8 | 3.9 ± 0.6 | 2.5 ± 1.0 | $F_{(1, 31)}$=0.03; p=0.86 | $F_{(1, 31)}$=3.42; p=0.075 | $F_{(1, 31)}$=0.009; p=0.93 |
| IL-2 | 1.7 ± 0.2 | 1.3 ± 0.3 | 2.1 ± 0.2 | 1.1 ± 0.3* | $F_{(1, 31)}$=0.16; p=0.69 | $F_{(1, 31)}$=11.14; p=0.002 | $F_{(1, 31)}$=1.83; p=0.19 |
| IL-3 | 0.9 ± 0.1 | 0.7 ± 0.1 | 1.0 ± 0.1 | 0.8 ± 0.2 | $F_{(1, 31)}$=0.32; p=0.58 | $F_{(1, 31)}$=1.51; p=0.23 | $F_{(1, 31)}$<0.001; p=0.98 |
| IL-5 | 14.1 ± 2.1 | 5.9 ± 1.1* | 12.4 ± 1.3 | 8.9 ± 1.9 | $F_{(1, 31)}$=0.17; p=0.68 | $F_{(1, 31)}$=14.35; p<0.001 | $F_{(1, 31)}$=2.34; p=0.14 |
| IL-6 | 3.1 ± 0.5 | 3.9 ± 1.3 | 3.5 ± 0.6 | 3.1 ± 1.0 | $F_{(1, 31)}$=0.046; p=0.83 | $F_{(1, 31)}$=0.023; p=0.88 | $F_{(1, 31)}$=0.5; p=0.49 |
| IL-10 | 4.8 ± 1.4 | 3.9 ± 1.5 | 3.4 ± 0.9 | 3.9 ± 1.9 | $F_{(1, 31)}$=0.24; p=0.63 | $F_{(1, 31)}$=0.027; p=0.87 | $F_{(1, 31)}$=0.25; p=0.63 |
| IL-12 | 6.2 ± 2.3 | 5.0 ± 1.7 | 20.9 ± 4.0# | 5.2 ± 2.4* | $F_{(1, 31)}$=8.44; p=0.007 | $F_{(1, 31)}$=10.85; p=0.003 | $F_{(1, 31)}$=7.97; p=0.009 |
| IL-15 | 40.2 ± 8.0 | 62.4 ± 25.5 | 49.7 ± 16.4 | 48.9 ± 14.4 | $F_{(1, 31)}$=0.016; p=0.9 | $F_{(1, 31)}$=0.44; p=0.51 | $F_{(1, 31)}$=0.51; p=0.48 |
| IL-17 | 1.8 ± 0.4 | 2.9 ± 0.8 | 3.4 ± 0.5 | 2.4 ± 1.0 | $F_{(1, 31)}$=0.66; p=0.42 | $F_{(1, 31)}$=0.0026; p=0.96 | $F_{(1, 31)}$=2.61; p=0.12 |
| CCL-2 | 17.9 ± 4.6 | 12.3 ± 4.8 | 9.3 ± 1.6 | 11.2 ± 5.4 | $F_{(1, 31)}$=1.42; p=0.24 | $F_{(1, 31)}$=0.22; p=0.64 | $F_{(1, 31)}$=0.87; p=0.36 |
| CCL-5 | 7.1 ± 0.6 | 6.0 ± 1.2 | 7.23 ± 1.2 | 5.4 ± 0.9 | $F_{(1, 31)}$=0.05; p=0.82 | $F_{(1, 31)}$=2.56; p=0.12 | $F_{(1, 31)}$=0.11; p=0.74 |
| IFN-γ | 2.6 ± 0.2 | 2.9 ± 0.3 | 5.4 ± 0.9# | 3.7 ± 0.6* | $F_{(1, 31)}$=10.97; p=0.03 | $F_{(1, 31)}$=1.96; p=0.17 | $F_{(1, 31)}$=3.66; p=0.066 |
| CXCL-1 | 105.6 ± 17.5 | 91.0 ± 33.3 | 83.9 ± 14 | 90.3 ± 20.2 | $F_{(1, 31)}$=0.28; p=0.6 | $F_{(1, 31)}$=0.038; p=0.85 | $F_{(1, 31)}$=0.25; p=0.62 |
| CXCL-5 | 368.4 ± 103.9 | 357.1 ± 155.5 | 346.6 ± 25.4 | 181.2 ± 48.4 | $F_{(1, 31)}$=0.67; p=0.42 | $F_{(1, 31)}$=0.017; p=0.9 | $F_{(1, 31)}$=0.003; p=0.96 |
| CXCL-9 | 207.4 ± 54.3 | 225.9 ± 53.9 | 346.6 ± 25.4# | 181.2 ± 48.4* | $F_{(1, 31)}$=1.15; p=0.29 | $F_{(1, 31)}$=2.79; p=0.11 | $F_{(1, 31)}$=4.37; p=0.046 |

behavioral impairments—are conserved in humans, and similar approaches can be implemented for clinical use.

## Methods
### Animals and TRF paradigm

All experimental procedures were approved by the UCLA Animal Research Committee and conformed to guidelines from the UCLA Division of Laboratory Animal Medicine (DLAM) and the National Institutes of Health (NIH). Wild-type (WT) (JAX ID: 000664) and *Fmr1* KO mice (*Fmr1*tm4Cgr) on the C57BL/6J background (JAX ID: 003025) were acquired from the Jackson laboratory (Bar Harbor, ME). These mutants have a neomycin resistance cassette replacing exon 5 of the fragile X mental retardation syndrome 1 (*Fmr1*) gene. The studies were carried out in male adult mice (3–5 months old) singly housed in light-tight ventilated cabinets in temperature- and humidity-controlled conditions, with ad

*Figure 8 continued*

spent in social interactions with the novel mouse in the fifth trial was increased to WT-like levels in the mutants on TRF. Paired *t*-tests were used to evaluate significant differences in the time spent interacting with the test mouse in the fourth (familiar mouse) and fifth (novel mouse) trials. *p < 0.05 indicates the significant time spent with the novel mouse compared to the familiar mouse. (**B**) Grooming was assessed in a novel arena in mice of each genotype (WT, *Fmr1* KO) under each feeding condition and the resulting data analyzed by two-way ANOVA followed by the Holm–Sidak's multiple comparisons test with feeding regimen and genotype as factors. *p < 0.05 indicates the significant difference within genotype – between diet regimens, and #p < 0.05 those between genotypes – same feeding regimen. (**C**) TRF did not alter the overall locomotion in the treated mice. See *Table 8*.

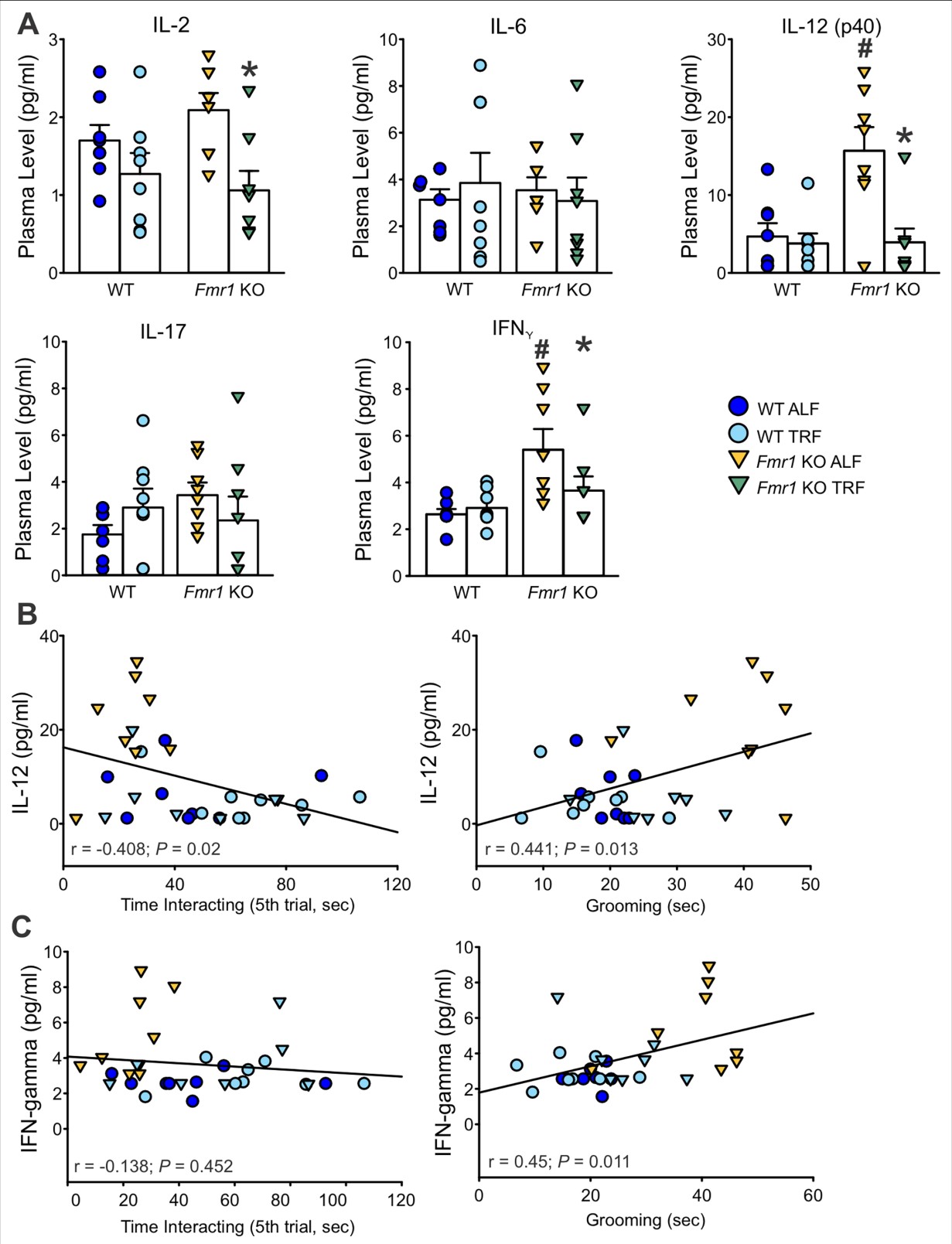

**Figure 9.** Time-restricted feeding (TRF) repristinates the levels IL-12 and IFN$_\gamma$ in the plasma of *Fmr1* KO mice to WT levels. (**A**) The levels of selected plasma pro-inflammatory markers are shown. The increase in IL-12 and IFN$_\gamma$ in the mutants is lowered by TRF to WT levels. The full list of the assayed makers is reported in *Table 9*. Data were analyzed with two-way ANOVA followed by the Holm–Sidak's multiple comparisons test with feeding regimen and genotype as factors. *p < 0.05 indicates the significant difference within genotype – between feeding regimen, and #p < 0.05 between genotypes –

*Figure 9 continued on next page*

**Table 10.** Correlation of the plasma levels of selected inflammatory markers with the severity of sleep disturbances and behavioral deficits.

Data from all four groups (WT ALF, WT TRF, Fmr1 KO ALF, Fmr1 KO TRF) were pooled and the Pearson correlation was applied. The correlation coefficients are reported; those significant are shown in bold and labeled with an asterisk. Alpha = 0.05.

| Measures | IL-12 (pg/ml) | IFN-γ (pg/ml) | CXCL-9 (pg/ml) |
|---|---|---|---|
| Rhythmic power (% V) | −0.34 | −0.15 | 0.072 |
| Activity in the light phase (ZT 0–3, a.u.) | 0.73* | 0.59* | 0.58 |
| Onset variability (min) | 0.19* | 0.69 | 0.01 |
| Sleep duration (min) | −0.48* | −0.48* | 0.34 |
| Sleep bout counts (#) | 0.26 | 0.16 | −0.16 |
| Avg. sleep bout length (min) | −0.36* | −0.24 | 0.25 |
| MAX sleep bout length (min) | −0.38* | −0.42* | 0.34 |
| Social recognition (trial 5, s) | −0.41* | −0.14 | 0.34 |
| Repetitive behavior (grooming, s) | 0.44* | 0.45* | 0.26 |

libitum access to food and water unless otherwise stated. Cages were equipped with either running wheel or infrared motion sensors to monitor the sleep/wake behavior.

TRF was conducted as previously described (*Wang et al., 2018*; *Whittaker et al., 2018*; *Whittaker et al., 2023*). Briefly, WT and *Fmr1* KO mice were singly housed and entrained to a 12:12 hr light/dark (LD) cycle (300 lx vs 0 lx, respectively) for a minimum of 2 weeks, then randomly assigned to one of two feeding conditions: food (standard chow) available ad libitum (ad lib) or available for 6 hr during the middle of the active phase from Zeitgeber Time (ZT) 15 to ZT 21. By definition, ZT0 and ZT12 are, respectively, the onsets of lights turning on and off. Scheduled feeding was achieved by manually adding and removing food from the mouse cages, and a careful examination was carried out to ensure no small food fragments were dropped and remained in the cages. Food consumption was measured by weighing food at the beginning and the end of the feeding cycles (ALF: 24 hr vs TRF: 6 hr). The ALF and TRF mice groups were held in these conditions for at least 2 weeks (14 days) plus 3 additional days to measure immobility-based sleep behavior (*Figure 2—figure supplement 1*).

This study used two different cohorts of mice for a total of 135 mice. The experiments shown in *Figures 1–6* were designed to compare the circadian phenotype of the *Fmr1* KO mice to the WT and were carried out between 2017 and 2020 before the Covid pandemic research shutdown. After the restart of research, new cohorts of mice were obtained from Jackson Laboratory and used to test the effects of scheduled feeding on the mutants' behavioral and circadian deficits (*Figures 7–9*).

## Immobility-based sleep behavior

Sleep behavior was assessed under a 12:12 hr LD cycle by video recording in combination with an automated mouse tracking analysis software system (Anymaze, Stoelting Co, Wood Dale, IL) as previously described (*Lee et al., 2018*; *Whittaker et al., 2018*). Sleep was defined as the periods when 95% of the animal remained immobile for at least 40 s or longer, a threshold shown to correlate with >99% accuracy to EEG-defined sleep (*Fisher et al., 2016*). For each animal, continuous recordings were performed over 5 days and data from two consecutive days with the fewest artifact-free epochs were averaged and used for further analysis. Data were exported in 1 min bins and analyzed separately for the light and dark phases. Total sleep time was determined by summing the immobility durations in the rest phase (ZT 0–12) or active phase (ZT 12–24). Sleep fragmentation was determined

*Figure 9 continued*

same feeding regimen. (**B, C**) Correlations between IL-12 or IFN-γ levels and social recognition or grooming behavior. IL-12 levels correlated with both behaviors. Data were analyzed using the Pearson Correlation, and the coefficients are reported in *Table 10*.

by the number of sleep bouts, which were operationally defined as episodes of continuous immobility with a sleep count greater than 3 per minute, persisting for at least 60 s.

## Locomotor cage activity rhythms

Methods used to characterize the sleep/wake cycles have been previously described (*Lee et al., 2018*; *Wang et al., 2018*; *Whittaker et al., 2018*). Single-housed WT and the *Fmr1* KO mice were habituated to a 12:12 hr LD cycle for 2 weeks to ensure entrainment to the appointed LD schedule before data collection. For the phenotypical comparisons between WT and *Fmr1* KO, cage activity was recorded with running wheel sensors for at least 14 days in LD, then, after the assessment of sleep behavior (3 days), the animals were released for 2 additional weeks into constant darkness (12:12 hr dark–dark, DD) to obtain measures of the endogenous rhythms (free-running activity; *Figure 2— figure supplement 1*). To investigate the effects of TRF, cage activity rhythms were monitored using overhead passive infrared motion sensors for 2 weeks (*Wang et al., 2018*; *Whittaker et al., 2018*).

Activity data were collected via the VitalView data recording system (Mini Mitter, Bend, OR) and 10 days of recordings were used to obtain the period (tau), the rhythmic power as well as the waveform presentations. The analysis was conducted using the El Temps (A. Diez-Nogura, Barcelona, Spain) and ClockLab (Actimetrics, Lafayette Instruments, Lafayette, IN) programs as previously described (*Lee et al., 2018*; *Wang et al., 2018*; *Whittaker et al., 2018*). The periodogram generated by the El Temps uses $\chi^2$ test with a threshold of 0.001 significance, from these the amplitude of the periodicities is calculated at the circadian harmonic to obtain the power of the rhythmicity. The rhythmic power, or percentage of variation (%V), provides a measure of the strength of the mouse's periodicity corrected for the activity amount and normalized to the percentage of variance derived from peak significance (p = 0.05). The amount of cage activity over a 24-hr period was averaged over 10 days and reported here as the arbitrary units (a.u.)/hr. Fragmentation and imprecision of the daily onset of sleep/wake cycles were determined using ClockLab. Fragmentation was determined by the number of activity bouts per day, with one bout counted when activity was separated by a gap of 21 min or more (maximum gap: 21 min; threshold 3 counts/min). Imprecision, a measure of cycle-to-cycle variability or onset variability, was determined by calculating the daily variability in the time of activity onset from a best-fit regression line drawn through 10 days of activity in both LD and DD conditions using the ClockLab program (Actimetrics, Wilmette, IL, United States).

## Photic regulation of circadian behavior

Cohorts of age-matched WT and *Fmr1* KO mice were single housed in cages with running wheels under a 12:12 hr LD cycle (300 lx, 4500 K) for 2 weeks and, after stable entrainment, exposed to four behavioral assays to test the photic regulation of their circadian system.

1. Photic suppression on nocturnal activity (negative light masking): mice were exposed to 1 hr of white light (300 lx, 4500 K) at ZT 14 to determine their response to light exposure during the dark period (masking of nocturnal activity). The animals' activity level, measured as the number of wheel revolutions, during the light pulse was compared to the level at the same hour (ZT 14–15) on the day before the exposure. Data are reported as percent of suppression = % [(rev during the 1 hr light exposure) – (rev during the dark baseline)]/(rev during the dark baseline).
2. Re-entrainment to a 6-hr advance of the LD cycle: the mice underwent a 6-hr phase advance. The activity levels were continuously monitored by the wheel-running system, and the activity onset was determined with VitalView. The ability of the animals to re-entrain was quantified by the difference between the new ZT 12 (lights-off) and the onset of the running activity for each subsequent recording day. A mouse was considered fully re-entrained when the activity onset aligned to the new lights-off time for the consecutive 5 days.
3. Skeleton photo period (SPP): the mice were transferred from the 12:12 hr LD cycle to an SPP consisting of 1hL:11hD:1L:11hD cycles. Mice were exposed to the 1 hr light (300 lx) at the beginning (ZT 0) and the end (ZT 11) of the original light phase. The mice were kept in this photic condition for at least 2 weeks and activity was recorded and analyzed to determine rhythm power, period (tau), and other basic activity rhythm parameters using ClockLab.
4. Light-induced phase shift of activity rhythms: Mice were released into DD for 10 days. The CT of their free-running activity rhythms was determined using VitalView and El Temps software. The time of activity onset under DD was defined as CT 12. On day 11, the mice were exposed to light (300 lx, 4500 K) at CT 16 for 15 min, and after the light pulse, they were held in DD for an

additional 10 days. The best-fit lines of the activity onsets of sleep/wake cycles before and after light exposure were measured and compared.

## Behavioral assays

All these tests were conducted under dim red light (<2 lx) during the dark (active) phase between 2 and 4 hr after lights off (ZT 14–16) to avoid sleep disruptions after the sleep and activity recordings had been completed (see *Figure 2—figure supplement 1*). Animals were habituated to the testing environment for at least 30 min prior to the behavioral assays.

The stereotypic behavior was assessed using two tests:

1. The marble burying test was used to evaluate repetitive digging behavior (*Thomas et al., 2009*). An array of 4 by 6 marbles was placed in the testing arena over a layer of 4 cm deep shavings. The testing mouse was introduced to the arena from the corner and its behavior recorded for 30 min. After that, the testing mouse was carefully removed from the arena with special precaution to not move the marbles and returned to the home cages. The number of marbles buried more than ⅔ of their area was counted, and these were cleaned with 70% ethanol before the next use. Time spent on digging was manually scored by an observer masked to the experimental groups, whilst the distance traveled was derived from the automatic mouse-tracking system (Anymaze software, Stoelting Co, Wood Dale, IL).
2. The grooming test was conducted as in our previous work (*Wang et al., 2020*). The mouse was introduced to the testing arena and its behavior recorded by a camcorder for 30 min. The time spent on self-grooming, defined as the cleaning, licking, or washing of the limbs, tail, and body surface areas, typically from a head to tail direction, but excluding bouts of scratching, was manually scored by an observer masked to the experimental groups. The distance traveled was derived from the automatic mouse-tracking system (Anymaze software).

Social behavior was assessed using two tests:

1. The three-chamber test was performed as previously described (*Wang et al., 2020*). The testing mice were allowed to freely explore an arena with three chambers where the central chamber remained empty. When being habituated to the three-chamber arena, both the WT and the mutants explored the arena evenly with no preference toward the left or the right chamber (left/right ratio: WT: 0.95 ± 0.14; p = 0.43 by paired *t*-test. *Fmr1* KO: 0.82 ± 0.1; p = 0.085 by paired *t*-test). The three-chamber test consisted of two parts: the first stage assessed the preference of social approach toward the stranger mouse, and the second stage assessed the ability of social discrimination of the testing mouse. In the first stage (social approach), an upturned metal-grid pencil cup was placed in the side chambers: one was left empty as the novel object (the object chamber), while a never-met stranger mouse matched by sex, age, and genotype with the testing mouse was placed in the second upturned cup (the social chamber). Thus, the testing mouse was tested for its preference between the object chamber and the social chamber in this first testing stage. In the second stage (social discrimination), the first stranger mouse and the cup remained the same, while a second stranger mouse matched by sex, age, and genotype with the testing mouse was placed in the second up-turned cup. In other words, the social chamber in the first stage became a familiar chamber in the second stage, and the object chamber of the first stage a novel chamber in the second stage. Thus, the preference between the familiar chamber and the novel chamber of the testing mouse was assessed in this second stage. Time spent in each chamber and the distance traveled were derived from the automatic mouse-tracking system (Anymaze software).
2. The five-trial social test was conducted as previously described (*Mineur et al., 2006*). The testing mouse was first habituated to the arena for 30 min and then introduced to a never-met stranger mouse for four trials. The testing mouse was allowed to explore and interact with the stranger mouse for 2 min in each trial, then it was introduced to a second stranger mouse during the fifth trial for 2 min. The resting interval between trials was 5 min. Active social behaviors such as physical contacts (e.g. crawling over, social grooming), nose-to-nose sniffing, and nose-to-anus from the testing mouse to the stranger mice were scored manually. Testing mice as well as stranger mice that showed aggressive behavior were withdrawn from the experiment.

## Retinal-SCN connectivity: injection, visualization, and analyses of the neuroanatomical tracer Cholera Toxin

WT and *Fmr1* KO mice (4 months old) received a bilateral injection of CholeraToxin (β subunit) conjugated to Alexa Fluor 555 Conjugate (catalog number: C34776; Invitrogen, Carlsbad, CA). Prior to the injections, the animals received a drop of a local ophthalmic anesthetic (Proparacaine HCl, 0.5%, Sandoz, Holzkirchen Germany) and an intraperitoneal (i.p.) injection of a non-steroidal anti-inflammatory drug (Carprofen, Zoetis, Parsippany-Troy Hills, NJ). The mice were then anesthetized with isoflurane, and a 30 G needle (BD PrecisionGlide Needle; Becton Dickinson, Franklin Lakes NJ) was inserted at a 45° angle into the sclera and the vitreous chamber, toward the base of the retina to allow leakage of vitreous humor. The Cholera Toxin (2 µg in 2 µl of sterile PBS) was injected into the vitreous chamber using a 32 G Hamilton syringe (Hamilton, Reno NV). The needle was left in place for 10 s before being retracted. Seventy-two hours after the injection, the mice were euthanized with isoflurane (30–32%) and transcardially perfused with phosphate-buffered saline (PBS, 0.1 M, pH 7.4) containing 4% (wt/vol) paraformaldehyde (PFA, Sigma). The brains were rapidly dissected out, post-fixed overnight in 4% PFA at 4°C, and cryoprotected in 15% sucrose. Sequential coronal sections (40–50 µm) containing the left and right SCN were mounted, and the cover slips applied with a drop of Vectashield-containing DAPI (4',6-diamidino-2-phenylindole; catalog number: H-1200; Vector Laboratories, Burlingame, CA). Sections were visualized on a Zeiss AxioImager M2 microscope equipped with an AxioCam MRm and the ApoTome imaging system, and images acquired with the Zeiss Zen software and a 10x objective to include both left and right SCN. Two methods of analyses were carried out on the images of five consecutive sections per animal containing the middle SCN (*Lee et al., 2018*). <u>First,</u> the relative intensity of the Cholera Toxin fluorescent processes was quantified in the whole SCN, both left and right separately, by scanning densitometry using the Fiji image processing package of the NIH ImageJ software (https://imagej.net). A single ROI of fixed size (575.99 µm × 399.9 µm, width x height) was used to measure the relative integrated density (mean gray values × area of the ROI) in all the images. The values from the left and right SCN were averaged per section, and 5 sections per animal were averaged to obtain one value per animal. <u>Second,</u> the retinal innervation of the SCN is strongest in the ventral aspect, where the retino-hypothalamic fibers reach the nuclei; hence, the distribution of the Cholera Toxin fluorescent signal was also obtained for each left and right ventral SCN separately in the same 4–5 consecutive sections per animal using the Profile Plot Analysis feature of ImageJ (*Lee et al., 2018*). Briefly, a rectangular box of fixed size (415.38 µm × 110.94 µm, width × height) to include the ventral part of the SCN was set for each side, and a column plot profile was generated whereby the *x*-axis represents the horizontal distance through the SCN (lateral to medial for the left and medial to lateral for the right; see *Figure 5—figure supplement 1*) and the *y*-axis represents the average pixel intensity per vertical line within the rectangular box. Subsequent processing of the resulting profiles was performed for left and right SCN images separately. To average the profiles of the 5 sections and obtain a single curve per animal, fifth-order polynomial curves were fit to best estimate the position of the intensity peak on the *x*-axis and, using this position, the original *y*-axis values were aligned and averaged arithmetically [1 profile per section (either left or right), 5 sections per animal]. Analyses were performed by two observers masked to the genotype of the animals. Data are shown as the average profile ± SD of 3 animals per genotype.

## Photic induction of cFos in the SCN and cFos-positive cell counting

A separate cohort of male WT and *Fmr1* KO mice (3–4 months old) was housed in DD conditions and exposed to light (300 lx, 4500 K, 15 min) at CT 16. Forty-five minutes later, the mice were euthanized with isoflurane (30–32%) and transcardially perfused with phosphate-buffered saline (PBS, 0.1 M, pH 7.4) containing 4% (wt/vol) PFA. The brains were rapidly dissected out, post-fixed overnight in 4% PFA at 4°C, and cryoprotected in 15% sucrose until further processing. Sequential coronal sections (40–50 µm), containing the middle SCN, were collected on a cryostat (Leica, Buffalo Grove, IL) and further processed for cFos immunofluorescence as previously described (*Wang et al., 2020*; *Wang et al., 2023*; *Longcore et al., 2024*). Briefly, free-floating coronal sections, paired on the rostral–caudal axis and containing both the left and right SCN, were blocked for 1 hr at room temperature (1% bovine serum albumin, 0.3% Triton X-100, 10% normal donkey serum in 1× PBS) and then incubated overnight at 4°C with a rabbit polyclonal antiserum against cFos (1:1000, RRID:AB_2247211; Cell Signaling) followed by a Cy3-conjugated donkey-anti-rabbit secondary antibody (Jackson

ImmunoResearch Laboratories, Bar Harbor, ME). Sections were mounted and coverslips applied with Vectashield mounting medium containing DAPI and visualized on a Zeiss AxioImager M2 microscope (Zeiss, Thornwood NY) equipped with a motorized stage, an AxioCam MRm, and the ApoTome imaging system. Z-Stack Images (35 images; 34 mm, 1.029 mm interval) of both the left and right middle SCN were acquired with a 20X objective using the Zeiss Zen digital imaging software, and two observers masked to the experimental groups performed the cell counting. The boundaries of the SCN were visualized using the DAPI nuclear staining and the cells immuno-positive for cFos counted with the aid of the Zen software tool 'marker' in three to five consecutive sections. The numbers obtained from the left and right SCN were averaged to obtain one value per section, and those from three to five sections averaged to obtain one value per animal and are presented as the mean ± standard deviation (SD) of 4 animals per genotype.

## Histomorphometrical analyses of the SCN

Photographs of DAPI-stained sections generated from the WT and *Fmr1* KO mice as described above were used to estimate the area, the perimeter, height, and width of the SCN as previously reported (*Li et al., 2015*; *Lee et al., 2018*). For each animal, the four measurements were performed in three consecutive sections containing the middle SCN and acquired with a 10X objective and the Zen software. Measurements (in µm) of both the left and right SCN were obtained with the auxilium of the AxioVision software (Zeiss, Pleasanton, CA, USA). Because the borders of the DAPI-defined SCN are somewhat arbitrary, measurements were performed independently by two observers masked to the genotype of the animals. The area of the SCN in the three sections was summed, whilst the perimeter, height, and width were averaged to obtain one value per side. No significant differences were found between the left and right SCN; therefore, the values of the left and right SCN were averaged to obtain one value per animal. Data are shown as the mean ± standard deviation (SD) of 6 animals per genotype.

## Blood sampling and measurements of plasma immune molecules

Blood was collected (~0.5 ml per animal) in the beginning of the light phase via cheek puncture into microvette tubes coated with EDTA (Sarstedt, Numbrecht, Germany; see *Figure 2—figure supplement 1*). Tubes were gently inverted a few times and placed immediately on wet ice. Within 1 hr following collection, samples were centrifuged at 2500 rpm for 15 min at 4°C and the plasma collected into prelabeled Eppendorf tubes (Fisher Scientific, Hampton, NH), and immediately stored at –80°C until further processing using the Luminex Multiplex Assay at the UCLA Immune Assessment Core (https://www.uclahealth.org/pathology/services-immunoassays).

## Statistical analyses

Data analyses were performed using SigmaPlot 14.5 (Grafiti LLC, Palo Alto, CA) or Prism 10 (GraphPad Software, La Jolla, CA). The impact of the loss of *Fmr1* on the waveforms of sleep/wake cycles was analyzed using repeated measures two-way analysis of variance (ANOVA) with time and genotype factors. While a two-way ANOVA with genotype and treatment as factors was used for the differences in sleep bouts between light phase and dark phase. The Holm–Sidak's multiple comparisons test was applied to determine significant differences among the groups. The effect of TRF on the activity waveforms was analyzed using a three-way ANOVA with genotype, feeding regimen, and time as factors followed by Holm–Sidak's multiple comparisons test. The datasets were examined for normality (Shapiro–Wilk test) and equal variance (Brown–Forsythe test); genotypic differences in the behavioral tests were determined by Student $t$-test or the Mann–Whitney test. Correlations between circadian/sleep parameters and other behaviors were examined by applying the Pearson correlation analysis. Normal distribution of the histomorphological datasets and the relative intensity of the Cholera Toxin fluorescent processes in the whole SCN were assessed using the Shapiro-Wilk test, and since these did not pass the normality test, a two-tailed Mann–Whitney test was employed to identify significant differences between groups. The effect of the loss of FMRP on the photic induction of cFos cells was assessed by one-way ANOVA followed by Bonferroni's multiple comparisons test. Values are reported as the mean ± standard error of the mean (SEM) or mean ± standard deviation (SD). Differences were determined significant if $p < 0.05$.

## Acknowledgements

Supported by National Institute of Child Health Development under award number: P50HD103557 (PIs S Bookheimer, H Kornblum).

## Additional information

### Funding

| Funder | Grant reference number | Author |
| --- | --- | --- |
| Eunice Kennedy Shriver National Institute of Child Health and Human Development | P50HD103557 | Christopher S Colwell |

The funders had no role in study design, data collection, and interpretation, or the decision to submit the work for publication.

### Author contributions

Huei-Bin Wang, Conceptualization, Data curation, Software, Formal analysis, Supervision, Investigation, Visualization, Methodology, Writing – original draft; Natalie E Smale, Data curation, Software, Investigation, Methodology; Sarah H Brown, Software, Investigation, Methodology; Sophia AMB Villanueva, David Zhou, Aly Mulji, Deap S Bhandal, Kyle Nguyen-Ngo, John R Harvey, Investigation, Methodology; Cristina A Ghiani, Conceptualization, Data curation, Software, Formal analysis, Supervision, Validation, Investigation, Visualization, Methodology, Writing – original draft, Project administration, Writing – review and editing; Christopher S Colwell, Conceptualization, Data curation, Software, Formal analysis, Supervision, Funding acquisition, Validation, Visualization, Writing – original draft, Project administration, Writing – review and editing

### Author ORCIDs

David Zhou ⬤ https://orcid.org/0000-0001-6066-3277
Cristina A Ghiani ⬤ https://orcid.org/0000-0002-9867-6185
Christopher S Colwell ⬤ https://orcid.org/0000-0002-1059-184X

### Ethics

All experimental procedures were approved by the UCLA Animal Research Committee (protocol ARC-2020-009) and conformed to guidelines from the UCLA Division of Laboratory Animal Medicine (DLAM) and the National Institutes of Health (NIH).

Reviewer #1 (Public review): https://doi.org/10.7554/eLife.104720.4.sa1
Reviewer #2 (Public review): https://doi.org/10.7554/eLife.104720.4.sa2
Author response https://doi.org/10.7554/eLife.104720.4.sa3

## Additional files

### Supplementary files

MDAR checklist

### Data availability

The raw data for Sleep and Activity rhythms, Behavioral assays, Cytokine measurements, light-induced cFos and profile plot analyses have been deposited in Dryad, University of California Curation Center (UC3), California Digital Library: https://doi.org/10.5061/dryad.bnzs7h4q1.

The following dataset was generated:

| Author(s) | Year | Dataset title | Dataset URL | Database and Identifier |
|---|---|---|---|---|
| Ghiani CA, Colwell CS | 2025 | Scheduled feeding improves behavioral outcomes and reduces inflammation in a mouse model of fragile X syndrome | https://doi.org/10.5061/dryad.bnzs7h4q1 | Dryad Digital Repository, 10.5061/dryad.bnzs7h4q1 |

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
