## [Editor Report · eLife Assessment]

This manuscript presents **solid** experimental data using Fmr1 knockout mice to explore the **fundamental** role of Fmr1 in sleep regulation. The study supports the hypothesis that scheduled feeding can improve circadian rhythm and behavior in a mouse model of Fragile X syndrome. These findings may offer new insights into neurodevelopmental disorders and their potential treatment strategies.

---

## [Referee Report · Reviewer #1 (Public review)]

The authors conducted a comprehensive investigation into sleep and circadian rhythm disturbances in Fmr1 knockout (KO) mice, a model for Fragile X Syndrome (FXS). They began by monitoring daily home cage behaviors to identify disruptions in sleep and circadian patterns, then assessed the mice's adaptability to altered light conditions through photic suppression and skeleton photoperiod experiments. To uncover potential mechanisms, they examined the connectivity between the retina and the suprachiasmatic nucleus. The study also included an analysis of social behavior deficits in the mutant mice and tested whether scheduled feeding could alleviate these issues. Notably, scheduled feeding not only improved sleep, circadian, and social behaviors but also normalized plasma cytokine levels. The manuscript is strengthened by its focus on a significant and underexplored area-sleep deficits in an FXS model-and by its robust experimental design, which integrates a variety of methodological approaches to provide a thorough understanding of the observed phenomena and potential therapeutic avenues.

---

## [Referee Report · Reviewer #2 (Public review)]

Summary:

In the present study, the authors, using a mouse model of Fragile X syndrome, explore the intriguing hypothesis that restricting food access over the daily schedule will improve sleep patterns and subsequently enhance behavioral capacities. By restricting food access from 12h to 6h over the nocturnal period (the active period for mice), they show, in these KO mice, an improvement in the sleep pattern accompanied by reduced systemic levels of inflammatory markers and improved behavior. These data, using a classical mouse model of neurodevelopmental disorder (NDD), suggest that modifying eating patterns might improve sleep quality, leading to reduced inflammation and enhanced cognitive/behavioral capacities in children with NDD.

Overall, the paper is well-written and easy to follow. The rationale of the study is generally well introduced. Data are globally sound. The interpretation is overall supported by the provided data.

---

## [Author Response]

The following is the authors’ response to the previous reviews

**Recommendations for the authors:**

**Reviewer #1 (Recommendations for the authors):**

Thank you for the extensive response to my comments and questions.

**Reviewer #2 (Recommendations for the authors):**
(1) The Fmr1/Fxr2 double KO mice are not well described in the Introduction.

We have changed the sentence in the introduction to clarify that in Zhang et al ., 2008 they used a mouse lacking both the Fmr1 gene and its paralog Fxr2.

(3) The Authors decided not to discuss the potential translation of the present study to human patients, despite their final conclusion statement.

The paragraph below has been added to the end of the discussion:

“Translational Implications”

The present findings support the view that circadian disruption is not merely a downstream consequence of disease processes but actively contributes to symptom expression. Hence, the possibility that interventions designed to reinforce circadian rhythms can hold therapeutic value for individuals with FXS and related neurodevelopmental conditions. Given that sleep and circadian dysfunction are detectable early in development and are predictive of more severe clinical phenotypes, circadian-based interventions may be particularly beneficial if applied during periods of heightened neural plasticity. Importantly, time-restricted feeding represents a relatively low-cost, non-invasive strategy that could be feasibly implemented in realworld settings. Further translational work is needed to evaluate whether the mechanistic links identified here—between circadian misalignment, immune dysregulation, and behavioral impairments—are conserved in humans, and similar approaches can be implemented for clinical use.